# IL-17D-induced inhibition of DDX5 expression in keratinocytes amplifies IL-36R-mediated skin inflammation

**Xinhui Ni** [1,9], **Yi Xu** [1,9], **Wang Wang** [1,9], **Baida Kong** [1], **Jian Ouyang** [1], **Jiwei Chen**[1], **Man Yan**[1], **Yawei Wu**[1], **Qi Chen**[1], **Xinxin Wang** [1], **Hongquan Li**[1], **Xiaoguang Gao** [1], **Hongquan Guo** [1], **Lian Cui** [2], **Zeyu Chen** [2], **Yuling Shi**[2,3], **Ronghui Zhu**[4], **Wei Li** [4], **Tieliu Shi**[1], **Lin-Fa Wang**[5,6], **Jinling Huang**[7], **Chen Dong** [7,8] and **Yuping Lai** [1]✉

Aberrant RNA splicing in keratinocytes drives inflammatory skin disorders. In the present study, we found that the RNA helicase DDX5 was downregulated in keratinocytes from the inflammatory skin lesions in patients with atopic dermatitis and psoriasis, and that mice with keratinocyte-specific deletion of *Ddx5* (*Ddx5^ΔKC^*) were more susceptible to cutaneous inflammation. Inhibition of DDX5 expression in keratinocytes was induced by the cytokine interleukin (IL)-17D through activation of the CD93–p38 MAPK–AKT–SMAD2/3 signaling pathway and led to pre-messenger RNA splicing events that favored the production of membrane-bound, intact IL-36 receptor (IL-36R) at the expense of soluble IL-36R (sIL-36R) and to the selective amplification of IL-36R-mediated inflammatory responses and cutaneous inflammation. Restoration of sIL-36R in *Ddx5^ΔKC^* mice with experimental atopic dermatitis or psoriasis suppressed skin inflammation and alleviated the disease phenotypes. These findings indicate that IL-17D modulation of DDX5 expression controls inflammation in keratinocytes during inflammatory skin diseases.

Alternative splicing contributes to transcriptomic and proteomic diversity. This process is tightly regulated in different tissues, cell types and differentiation stages, but its dysregulation can drive tumorigenesis[1,2] or immune-related diseases[3]. Atopic dermatitis (AD) and psoriasis are clinically independent inflammatory skin diseases[4]. AD is strongly driven by helper type 2 T cells (T_H2 cells) and is associated with IL-4 and IL-13 overproduction, whereas psoriasis is largely driven by T_H17 cells

and associated with IL-17 activation[5]. Although an abnormal keratinocyte response to T cell-derived cytokines is intrinsic to the pathology of AD and psoriasis, whether they share common mechanisms that regulate keratinocyte inflammation remains unclear.

Deep RNA-sequencing (RNA-seq) and high-throughput, genome-wide transcriptome analyses of skin from healthy controls and patients with AD or psoriasis show that AD and psoriasis have a similar

[1]Shanghai Frontiers Science Center of Genome Editing and Cell Therapy, Shanghai Key Laboratory of Regulatory Biology, School of Life Sciences, East China Normal University, Shanghai, China. [2]Department of Dermatology, Shanghai Tenth People's Hospital, School of Medicine, Tongji University, Shanghai, China. [3]Shanghai Skin Disease Hospital, School of Medicine, Tongji University, Shanghai, China. [4]Department of Dermatology, Huashan Hospital, Fudan University, Shanghai, China. [5]Emerging Infectious Diseases Program, Duke–NUS Graduate Medical School, Singapore, Singapore. [6]Singhealth Duke–NUS Global Health Institute, Singapore, Singapore. [7]Institute for Immunology and School of Medicine, Tsinghua University, Beijing, China. [8]Shanghai Immune Therapy Institute, Shanghai Jiaotong University School of Medicine-affiliated Renji Hospital, Shanghai, China. [9]These authors contributed equally: Xinhui Ni, Yi Xu, Wang Wang. ✉e-mail: yplai@bio.ecnu.edu.cn

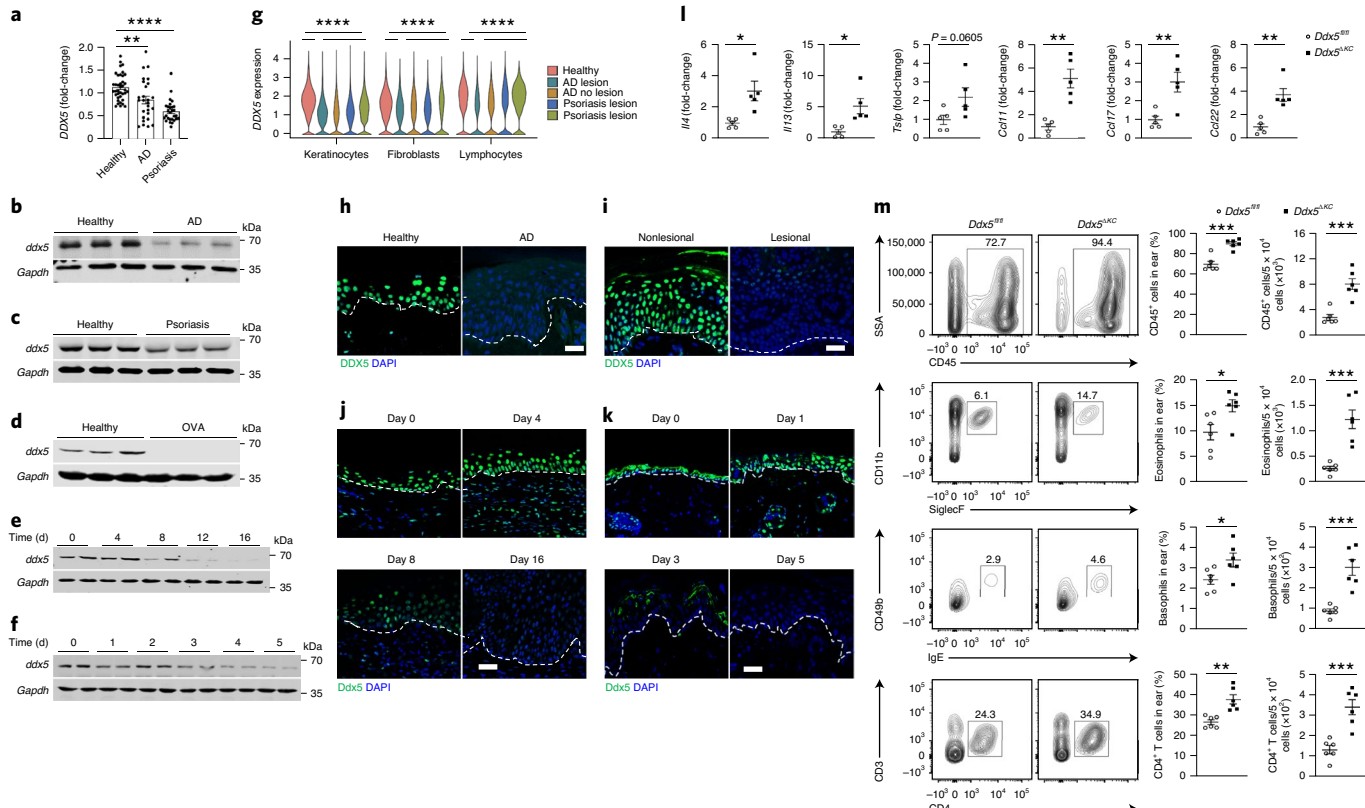

**Fig. 1 | DDX5 defect in epidermal keratinocytes drives skin inflammation during AD and psoriasis. a**, Expression of *DDX5* mRNA reanalyzed from a publicly available (GEO database, accession no. GSE121212) RNA-seq analysis of 38 healthy controls and patients with AD (*n* = 27) or psoriasis (*n* = 28)[6]. Data were normalized to the gene *GAPDH*. **b**,**c**, Immunoblot of DDX5 in healthy skin (*n* = 3) and lesional skin collected from three patients with AD (**b**) or lesional skin from three patients with psoriasis (**c**). **d**–**f**, Immunoblot of Ddx5 in skin extracts from wild-type mice topically treated with OVA patches on dorsal skin for 7 d consecutively after two intraperitoneal injections of OVA at 1-week intervals (**d**), MC903 for 15 d consecutively (**e**) or IMQ for 5 d consecutively (**f**). **g**, Expression of *DDX5* mRNA in keratinocytes, fibroblasts and lymphocytes in scRNA-seq data from the skin of five healthy adults and patients with AD (*n* = 4) and psoriasis

(*n* = 3). **h**–**k**, Immunofluorescence analysis of DDX5+ cells in healthy skin (*n* = 3) and lesional skin from three patients with AD (**h**), nonlesional and lesional skin from three patients with psoriasis (**i**) or lesional skin from MC903-treated (**j**) or IMQ-treated (**k**) wild-type mice at the indicated days. Scale bars, 25 μm. The dotted lines indicate the edge between the epidermis and dermis. **l**, RT–qPCR of indicated genes in lesional ear skin from MC903-treated *Ddx5*^fl/fl^ and *Ddx5*^ΔKC^ (*n* = 5) mice. **m**, Flow cytometry of CD45+ cells, CD11b+SiglecF+ eosinophils, CD49b+IgE+ basophils and CD3+CD4+ T cells in ears from MC903-treated *Ddx5*^fl/fl^ and *Ddx5*^ΔKC^ mice (*n* = 6). *$P < 0.05$, **$P < 0.01$, ***$P < 0.001$ and ****$P < 0.0001$. The *P* values were determined by one-way ANOVA (**a** and **g**) or unpaired, two-tailed Student's *t*-test (**l** and **m**). Data are presented as mean ± s.e.m.

profile of alternatively spliced transcripts in skin lesions[6–8]. Analysis of splicing signatures indicates that multiple candidate-splicing factors, including HNRNPA1, U2AF1 and DDX5, might be responsible for RNA-splicing changes in AD and psoriasis[6,7]. Among these splicing factors, the DEAD (AspGluAlaAsp) box (DDX), RNA helicase DDX5 regulates diverse aspects of RNA biogenesis by unwinding or destabilizing the stem–loop structure of genes to facilitate splicing factors binding to splice sites[9] and has been associated with disease in humans[10]. Downregulation or depletion of DDX5 leads to male infertility by changing the alternative splicing pattern of multiple genes during spermatogenesis[11], whereas high expression of DDX5 drives tumor development by modulating the alternative splicing of H-Ras or mH2A1 in breast cancer[12]. DDX5 also facilitates vesicular stomatitis virus propagation by promoting RNA decay of antiviral transcripts and nuclear export of transcripts DHX58, p65 and IKKγ in mouse embryo fibroblasts[13]. DDX5 has been predicted to be a shared biological factor against viral replication in inverse, erythrodermic and chronic plaque psoriasis[14] and genetic variants at the DDX5 locus predispose individuals to asthma, an atopic disease associated with AD[15,16].

The response of keratinocytes to the IL-17 family of cytokines, such as IL-17A, IL-17C, IL-17E (also named IL-25) and IL-17F, contributes

to the cycle of inflammation and cellular proliferation that results in epidermal hyperproliferation and lesion formation in AD and psoriasis[17–19]. IL-17A activates the IL-17RA–IL-17RC complex or IL-17RC–IL-17RD complex to induce keratinocyte proliferation and the expression of chemokines, such as CXCL1 and CCL20, that recruit neutrophils, T_H17 cells or γδ T cells into skin lesions in psoriasis[17,20]. IL-17E activates IL-17RB–STAT3 signaling in keratinocytes to drive skin inflammation in psoriasis[18] or the IL-17RA–IL-17RB complex to induce keratinocyte differentiation and the expression of IL-4 and IL-13 in AD[21]. IL-17C and IL-17F also induce keratinocyte proliferation and inflammatory cytokine production in psoriasis[22]. IL-17D, the least understood member of IL-17 family, regulates intestinal homeostasis through its receptor, CD93 (ref. [23]), but the role of IL-17D in skin inflammation is completely unknown.

To address whether dysregulation of DDX5 function or expression had a role in propagating inflammation in the skin, we analyzed the expression profile of DDX5 in AD and psoriasis and generated mice with conditional knockout of *Ddx5* in keratinocytes. We show that inhibition of DDX5 expression in keratinocytes downstream of IL-17D signaling amplified skin inflammation by controlling the pre-mRNA splicing of IL-36R, which in turn increased IL-36R expression and amplified the IL-36R-mediated skin inflammation.

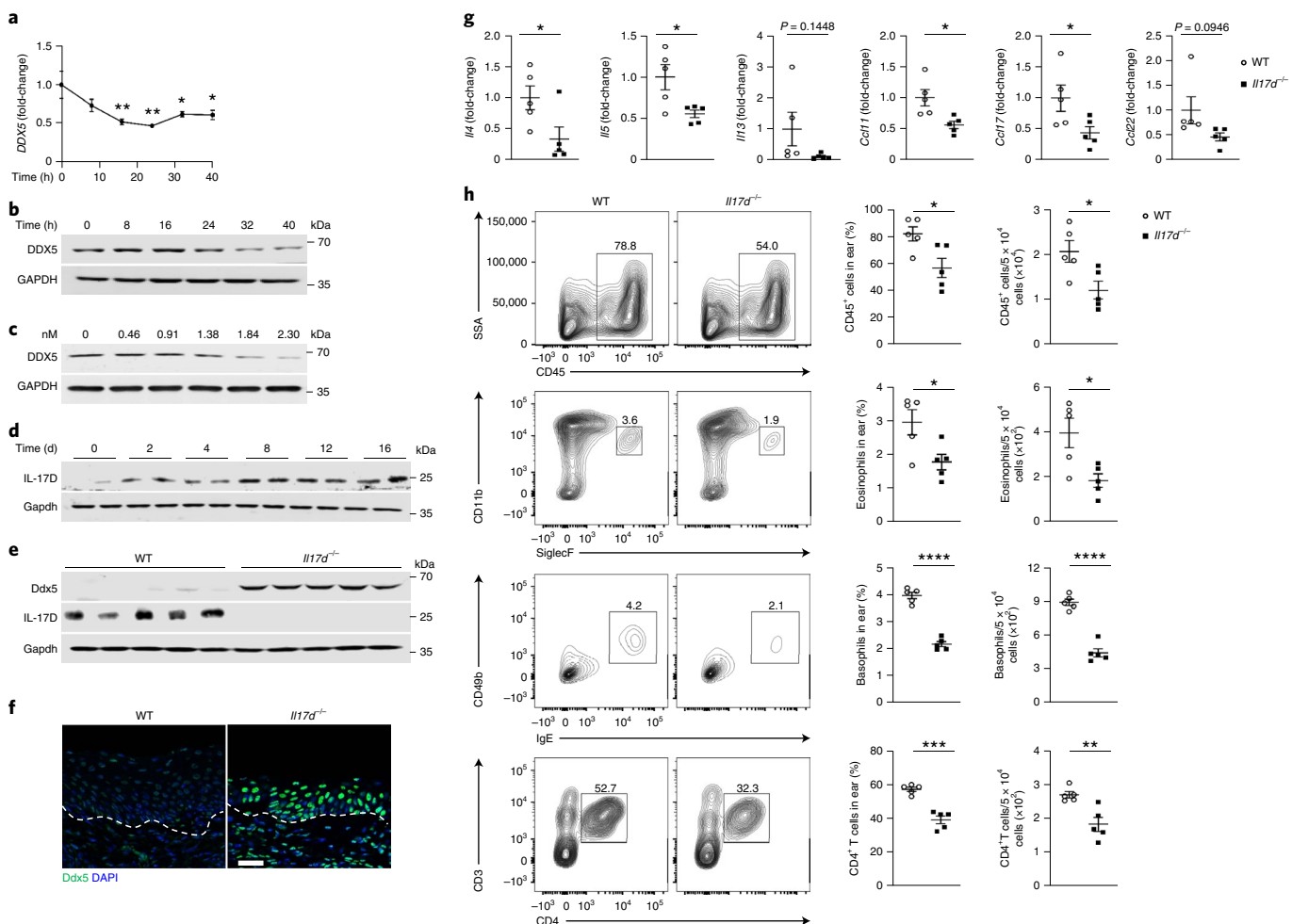

**Fig. 2 | IL-17D inhibits keratinocyte DDX5 to drive AD. a**, RT−qPCR of *DDX5* in NHEKs treated with 2.3 nM IL-17D for 0-40 h. **b**,**c**, Immunoblot of DDX5 in NHEKs treated with 2.3 nM IL-17D for 0-40 h (**b**) or 0−2.3 nM IL-17D for 24 h (**c**). **d**, Immunoblot of IL-17D in lesional skin from MC903-treated wild-type mice at day 0 to day 16. **e**, Immunoblot of Ddx5 and IL-17D in lesional skin from MC903-treated wild-type (WT) and *Il17d*⁻/⁻ mice. **f**, Immunofluorescence analysis of Ddx5⁺ cells in lesional skin from MC903-treated wild-type and *Il17d*⁻/⁻ mice (*n* = 5). Scale bars, 25 μm. The dotted lines indicate the edge between the epidermis and dermis. **g**, RT−qPCR of *Il4*, *Il5*, *Il13*, *Ccl11*, *Ccl17* and *Ccl22* in lesional skin from MC903-treated wild-type and *Il17d*⁻/⁻ mice (*n* = 5). **h**, Flow cytometry of CD45⁺ cells, CD11b⁺ SiglecF⁺ eosinophils, CD49b⁺IgE⁺ basophils and CD3⁺CD4⁺ T cells in ears from MC903-treated wild-type and *Il17d*⁻/⁻ mice (*n* = 5). *P < 0.05, **P < 0.01, ***P < 0.001 and ****P < 0.0001. The P values were determined by one-way ANOVA (**a**) or unpaired, two-tailed Student's *t*-test (**g** and **h**). Data are presented as mean ± s.e.m.

## Results

### DDX5 is reduced in keratinocytes and drives skin inflammation

Gene ontology (GO) and the *Kyoto Encyclopedia of Genes and Genomes* (KEGG) pathway enrichment analyses of overlapping differentially expressed genes (DEGs) from a publicly available RNA-seq dataset (Gene Expression Omnibus (GEO) database: accession no. GSE121212) from healthy controls and patients with AD or psoriasis[6] showed a similar RNA-splicing pattern, based on the presence of shared spliceosome complexes in the skin lesions of patients with AD and psoriasis (Extended Data Fig. 1a). Based on quantitative reverse transcription PCR (RT−qPCR), the mRNA for *DDX5*, a component of spliceosome complexes[9], but not all other tested splicing factors, including *CDC5L*, *PRPF18*, *DHX8*, *DHX15*, *HNRNPA1* and *SRSF1*, was significantly decreased in whole skin lesions from patients with AD and psoriasis compared with healthy controls (Fig. 1a and Extended Data Fig. 1b). Based on immunoblotting and immunofluorescence staining, expression of DDX5 protein was reduced in whole skin lesions from patients with AD and psoriasis (Fig. 1b,c) and slightly increased in epidermal tissues from skin carcinomas (Extended Data Fig. 1c), compared with healthy skin.

The mRNA and protein expression of Ddx5 were progressively reduced in whole lesional skin from wild-type mice treated with MC903 (calcipotriol), a vitamin D₃ analog, by painting the ears for 15 d consecutively or placing ovalbumin (OVA) patches on dorsal skin for 7 d consecutively after two intraperitoneal injections of OVA at 1-week intervals to induce AD, or imiquimod (IMQ), a toll-like receptor 7 agonist, painting on dorsal skin for 5 d consecutively to induce psoriasis, compared with skin from untreated wild-type mice (Fig. 1d−f and Extended Data Fig. 1d,e). Reanalysis of single-cell RNA-seq (scRNA-seq) datasets of skin from healthy adults and patients with AD and psoriasis[24] indicated that *DDX5* mRNA was downregulated in keratinocytes, fibroblasts and lymphocytes from lesions or nonlesional skin in patients with AD and psoriasis (Fig. 1g). Immunofluorescence staining indicated that DDX5 protein was highly expressed in epidermal keratinocytes from healthy skin of humans and mice (Fig. 1h−k), but was almost undetectable in whole lesional skin from patients with AD and psoriasis and MC903- or IMQ-treated mice (Fig. 1h−k), indicating a reduction in keratinocyte DDX5 in inflamed skin in humans and mice.

To determine the role of keratinocyte DDX5 in skin inflammation, we generated mice with a specific ablation of *Ddx5* in keratinocytes by

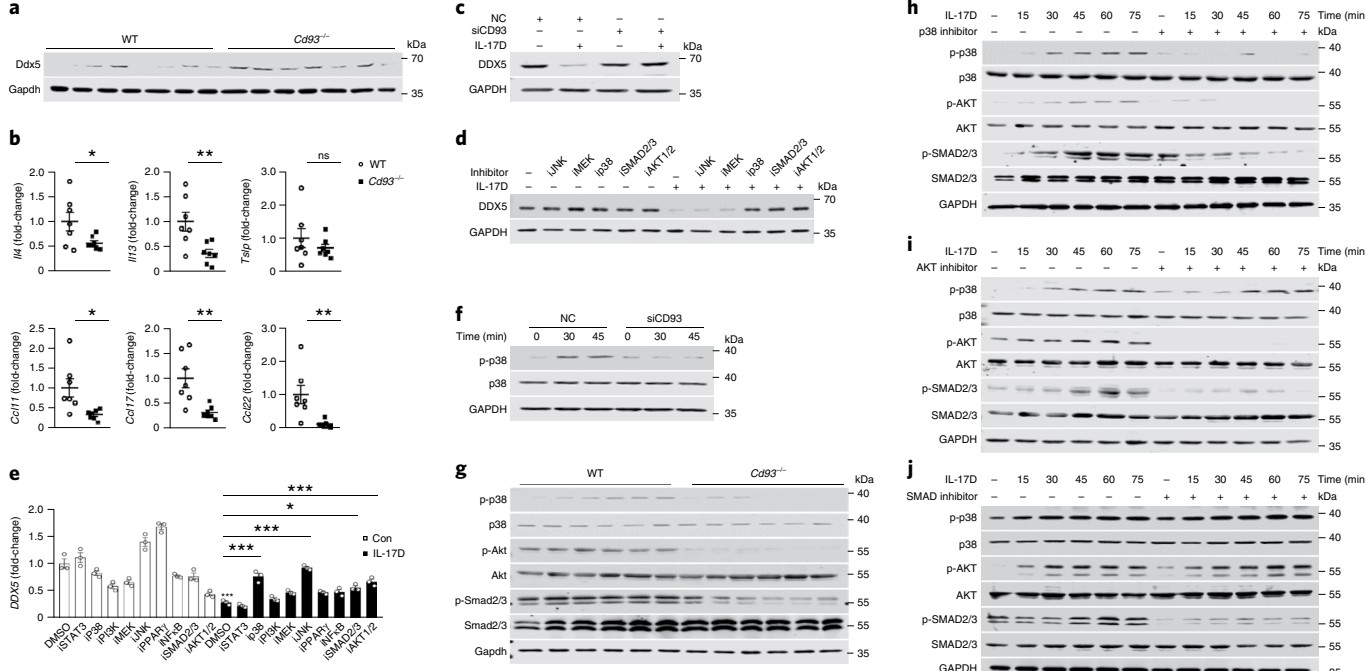

**Fig. 3 | IL-17D activates the CD93–p38 MAPK–AKT–SMAD2/3 signaling pathway to inhibit DDX5. a**, Immunoblot of Ddx5 in lesional skin from MC903-treated wild-type and *Cd93*[−/−] mice. **b**, RT–qPCR of *Il4*, *Il13*, *Tslp*, *Ccl11*, *Ccl17* and *Ccl22* in lesional skin from MC903-treated wild-type and *Cd93*[−/−] mice (*n* = 7). **c**, Immunoblot of DDX5 in NHEKs treated with 2.3 nM IL-17D 24 h after transduction of CD93-targeted siRNAs. **d**,**e**, Immunoblot (**d**) or RT–qPCR (**e**) of DDX5 in NHEKs 24 h after treatment with the STAT3 inhibitor S3I201, p38 MAPK inhibitor SB202190, phosphoinositide 3-kinase (PI3K) inhibitor LY294002, MEK inhibitor PD98059, JNK inhibitor SP600125, peroxisome proliferator-activated receptor-γ (PPARγ) inhibitor GW9662, nuclear factor κ-light chain enhancer of activated B cells (NFκB) inhibitor PDTC, SMAD2/3 inhibitor SB431542 and

AKT1/2 kinase inhibitor with or without 2.3 nM IL-17D. Con, Control; DMSO, Dimethylsulfoxide. **f**, Immunoblot of phosphorylated p38 MAPK and p38 MAPK in NHEKs transduced with *CD93*-targeted siRNA and treated with 2.3 nM IL-17D for 0–45 min after 24-h CD93 silencing. **g**, Immunoblot of p38 Mapk, Akt, Smad2/3 and their phosphorylation in lesional skin from MC903-treated wild-type and *Cd93*[−/−] mice. **h**–**j**, Immunoblot of p38 MAPK, AKT, SMAD2/3 and their phosphorylation in NHEKs treated with SB202190 (**h**) or AKT1/2 kinase inhibitor (**i**) or SB431542 (**j**) in the presence of 2.3 nM IL-17D for 0–75 min. [*]*P* < 0.05, [**]*P* < 0.01 and [***]*P* < 0.001. NS, not significant. *P* values were determined by unpaired, two-tailed Student's *t*-test (**b**) or two-way ANOVA (**e**). Data are presented as mean ± s.e.m.

---

crossing *Ddx5*[fl/fl] mice with K14Cre transgenic mice, hereafter referred to as *Ddx5*[ΔKC] mice. Compared with *Ddx5*[fl/fl] littermates, *Ddx5*[ΔKC] mice had greatly thickened scaling and more patches and earlier (day 16 compared with day 8 postinitiation of MC903) emergence of dryness and spongiosis on the ear skin (Extended Data Fig. 2a,b), which resembled clinical features in patients with AD[25]. The expression of *Il4, Il13, Ccl11, Ccl17* and *Ccl22* mRNA was increased two- to fourfold (Fig. 1l), whereas the expression of *Tslp* mRNA was not changed (Fig. 1l), in the skin lesions of *Ddx5*[ΔKC] mice compared with *Ddx5*[fl/fl] mice at day 16 post-MC903 administration. The percentages and absolute numbers of CD45[+] immune cells, including CD11b[+]SiglecF[+] eosinophils, CD49b[+]IgE[+] basophils and CD3[+]CD4[+] T cells, were increased in the ear skin of MC903-treated *Ddx5*[ΔKC] mice compared with *Ddx5*[fl/fl] littermates (Fig. 1m), indicating more severe inflammation. *Ddx5*[ΔKC] mice also had more thickened plaques and increased acanthosis on the dorsal skin on day 5 post-IMQ administration compared with *Ddx5*[fl/fl] littermates (Extended Data Fig. 2c,d). Moreover, the expression of *Il23, Il17a, Ccl20, Cxcl1, Cxcl2* and *S100a7* (Extended Data Fig. 2e), which is linked to the pathogenesis of psoriasis[26], and the infiltration of CD45[+] immune cells, including CD11b[+]Ly6G[+] neutrophils, CD45[+]MHCII[+] antigen-presenting cells (APCs) and γδTCR[+] T cells (Extended Data Fig. 2f), were increased in whole skin lesions in IMQ-treated *Ddx5*[ΔKC] mice compared with *Ddx5*[fl/fl] littermates. Thus, DDX5 deficiency in keratinocytes drove cutaneous inflammation.

## DDX5 expression is inhibited by IL-17D in keratinocytes

To investigate the factors that inhibited DDX5 expression in keratinocytes during inflammation we treated neonatal human epidermal

keratinocytes (NHEKs) with 11 inflammatory cytokines, including IL-17A, IL-17F, tumor necrosis factor (TNF), IL-1β, IL-4 and IL-36γ. Among all those tested, only IL-17D reduced the expression of DDX5 mRNA and protein in a time- and dose-dependent manner (Fig. 2a–c and Extended Data Fig. 3a,b). Production of IL-17D steadily increased in the skin of MC903- or IMQ-treated wild-type mice compared with untreated mice (Fig. 2d and Extended Data Fig. 4a). Expression of Ddx5 in whole lesional skin from MC903- or IMQ-treated *Il17d*[−/−] mice was higher compared with wild-type counterparts and was similar to that in healthy skin from wild-type mice (Fig. 2e,f and Extended Data Fig. 4b,c). MC903- or IMQ-treated *Il17d*[−/−] mice exhibited fewer plaques in the ear skin (Extended Data Fig. 4d), had reduced expression of *Il4, Il5, Il13, Ccl11, Ccl17* and *Ccl22* (Fig. 2g) and decreased infiltration of CD45[+] immune cells, including CD11b[+]SiglecF[+] eosinophils, CD49b[+]IgE[+] basophils and CD3[+]CD4[+] T cells (Fig. 2h and Extended Data Fig. 4e), in lesions compared with wild-type counterparts. These data suggested that IL-17D inhibited the expression of DDX5 in keratinocytes during AD and psoriasis.

## IL-17D activates CD93-mediated signaling to inhibit DDX5

CD93, a type I transmembrane protein with an amino-terminal C-type lectin-like domain has been identified as a receptor for IL-17D[23]. Ddx5 was increased in the whole skin lesions of MC903- or IMQ-treated *Cd93*[−/−] mice compared with their wild-type counterparts (Fig. 3a and Extended Data Fig. 5a), along with fewer plaques on the ears or dorsal skin (Extended Data Fig. 5b) and reduced expression of *Il4, Il13, Ccl11, Ccl17* and *Ccl22* (Fig. 3b) or *Cxcl1, Cxcl2, Ccl20, Il23, Il17a* and *Il36γ*

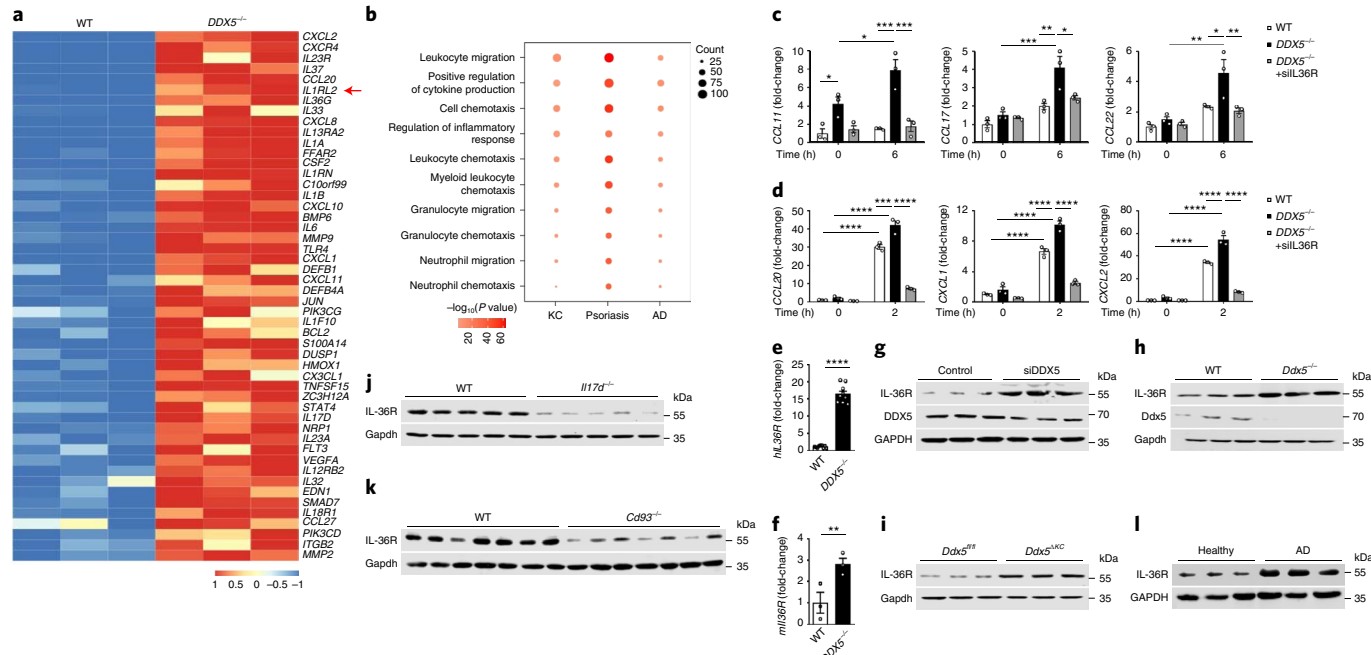

**Fig. 4 | DDX5 regulates the expression of IL-36R in keratinocytes and AD. a**, Heatmap showing the top 50 upregulated genes in wild-type and *DDX5*$^{-/-}$ HaCaT cells (*n* = 3) stimulated with 100 ng ml$^{-1}$ of IL-36γ for 4 h by RNA-seq analysis. The red arrow points to *IL1RL2* (IL-36R). **b**, The top ten categories from GO enrichment analysis of RNA-seq resulting in **a** and the whole lesional skin from MC903-treated *Ddx5*$^{ΔKC}$ (*n* = 4) and *Ddx5*$^{fl/fl}$ mice (*n* = 3) or IMQ-treated *Ddx5*$^{ΔKC}$ (*n* = 3) and *Ddx5*$^{fl/fl}$ mice (*n* = 3). **c,d**, RT–qPCR of *CCL11*, *CCL17* and *CCL22* (**c**) or *CCL20*, *CXCL1* and *CXCL2* (**d**) in wild-type and *DDX5*$^{-/-}$ HaCaT cells stimulated with 100 ng ml$^{-1}$ of IL-36γ for 6 h (**c**) or 2 h (**d**) before and after transduction of *IL36R*-targeting siRNA. **e,f**, RT–qPCR of human *IL36R* in wild-type and *DDX5*$^{-/-}$ HaCaT cells treated with 100 ng ml$^{-1}$ of human IL-36γ for 4 h (**e**) or murine *Il36R* in wild-type and *Ddx5*$^{ΔKC}$ murine keratinocytes treated with 100 ng ml$^{-1}$ of murine IL-36γ for 10 h (**f**). **g,h**, Immunoblot of IL-36R and DDX5 in NHEKs transduced with siRNA targeting *DDX5* (**g**) or in murine keratinocytes isolated from *Ddx5*$^{fl/fl}$ and *Ddx5*$^{ΔKC}$ mice (**h**). **i**–**k**, Immunoblot of IL-36R in lesional skin from MC903-treated *Ddx5*$^{fl/fl}$ and *Ddx5*$^{ΔKC}$ mice (**i**), wild-type and *Il17d*$^{-/-}$ mice (**j**) or wild-type and *Cd93*$^{-/-}$ mice (**k**). KC, keratinocytes. **l**, Immunoblot of IL-36R in healthy skin and lesional skin from patients with AD. *$P$ < 0.05, **$P$ < 0.01, ***$P$ < 0.001 and ****$P$ < 0.0001. The *P* values were analyzed by two-way ANOVA (**c** and **d**) or unpaired, two-tailed Student's *t*-test (**e** and **f**). Data are presented as mean ± s.e.m.

(Extended Data Fig. 5c). Silencing of CD93 in NHEKs by small interfering RNAs (siRNAs) targeting *CD93* (Fig. 3c and Extended Data Fig. 5d), and inhibition of p38 MAPK (mitogen-activated protein kinase), AKT1/2 and SMAD2/3 with SB202190, AKT1/2 kinase inhibitor and SB431542, respectively (Fig. 3d,e), abrogated the reduction in DDX5 expression in IL-17D-stimulated NHEKs compared with dimethylsulfoxide-treated NHEK cells. Treatment with IL-17D induced the phosphorylation of p38 MAPK, AKT and SMAD2/3 in NHEKs or whole lesional skin from MC903- or IMQ-treated wild-type mice, but this effect was lost in CD93-silenced NHEKs or whole lesional skin from MC903- or IMQ-treated *Cd93*$^{-/-}$ mice (Fig. 3f,g and Extended Data Fig. 5e). Moreover, the p38 MAPK inhibitor SB202190 dampened IL-17D-induced phosphorylation of AKT and SMAD2/3, whereas the AKT1/2 kinase inhibitor and SMAD2/3 inhibitor SB431542 did not block the IL-17D-induced phosphorylation of p38 MAPK in NHEKs (Fig. 3h–j). The AKT1/2 kinase inhibitor prevented IL-17D-induced phosphorylation of SMAD2/3, but the SMAD2/3 inhibitor had no effect on the IL-17D-induced phosphorylation of AKT1/2 (Fig. 3i,j). These data indicated that IL-17D activated a p38 MAPK–AKT–SMAD2/3 signaling pathway downstream of CD93 to inhibit expression of DDX5 in keratinocytes.

## DDX5 deficiency amplifies IL-36–IL-36R-mediated inflammation

AD and psoriasis share multiple drivers of epidermal hyperplasia and inflammation, such as IL-17A, TNF, IL-17E, IL-36 and IL-4 (refs. [5,18,27–29]). To test whether DDX5 regulated the keratinocyte response to these cytokines, we treated wild-type and *DDX5*$^{-/-}$ immortalized human keratinocytes (HaCaT cell line) with IL-17A, IL-36γ, TNF, IL-25 and IL-4. The expression of multiple chemokines, including *CCL20*,

*CXCL1*, *CXCL2*, *CXCL6*, *CCL3*, *CCL17* and *CCL22*, was increased in *DDX5*$^{-/-}$ HaCaT cells 2 h after stimulation with IL-36γ, whereas the expression of some chemokines, such as *CCL20* and *CXCL2*, was slightly increased in *DDX5*$^{-/-}$ HaCaT cells treated with TNF or IL-25 (Extended Data Fig. 6a–e), suggesting that *DDX5* deficiency amplified an IL-36γ-mediated inflammatory response in keratinocytes.

Next, we performed high-throughput RNA-seq in wild-type and *DDX5*$^{-/-}$ HaCaT cells 4 h post-IL-36γ stimulation or in lesional skin from MC903- or IMQ-treated *Ddx5*$^{ΔKC}$ and *Ddx5*$^{fl/fl}$ mice. Global transcriptional profiling showed that the expression of multiple inflammatory mediators such as *IL-1β*, *CXCL1* and *CXCL2* was increased in *DDX5*$^{-/-}$ HaCaT cells or in lesions of MC903- or IMQ-treated *Ddx5*$^{ΔKC}$ mice compared with IL-36γ-stimulated, wild-type HaCaT cells or lesions from MC903- or IMQ-treated *Ddx5*$^{fl/fl}$ mice (Fig. 4a and Extended Data Fig. 6f,g). GO enrichment analysis of three RNA-seq results from IL-36γ-stimulated wild-type and *DDX5*$^{-/-}$ HaCaT cells or skin lesions from MC903- or IMQ-treated *Ddx5*$^{ΔKC}$ and *Ddx5*$^{fl/fl}$ mice indicated that the DDX5-dependent changes in gene expression mostly affected leukocyte migration, positive regulation of cytokine production and cell chemotaxis (Fig. 4b). RT–qPCR confirmed that the expression of mRNA for *CCL11* (3.6-fold), *CCL17* (2.6-fold) and *CCL22* (3.1-fold), which attract eosinophils and $T_H2$ cells to AD skin lesions[30], and the expression of mRNA for *CCL20* (40.5-fold), *CXCL1* (8.5-fold) and *CXCL2* (51.2-fold), which recruit γδT cells and neutrophils to psoriatic lesions[31], were significantly increased in *DDX5*$^{-/-}$ HaCaT cells compared with wild-type HaCaT cells in response to IL-36γ (Fig. 4c,d). The siRNAs targeting *IL1RL2*, which encodes IL-36R, in IL-36γ-stimulated *DDX5*$^{-/-}$ HaCaT cells decreased the expression of IL-36γ-induced *CCL11*, *CCL17*, *CCL22*, *CCL20*, *CXCL1* and *CXCL2*

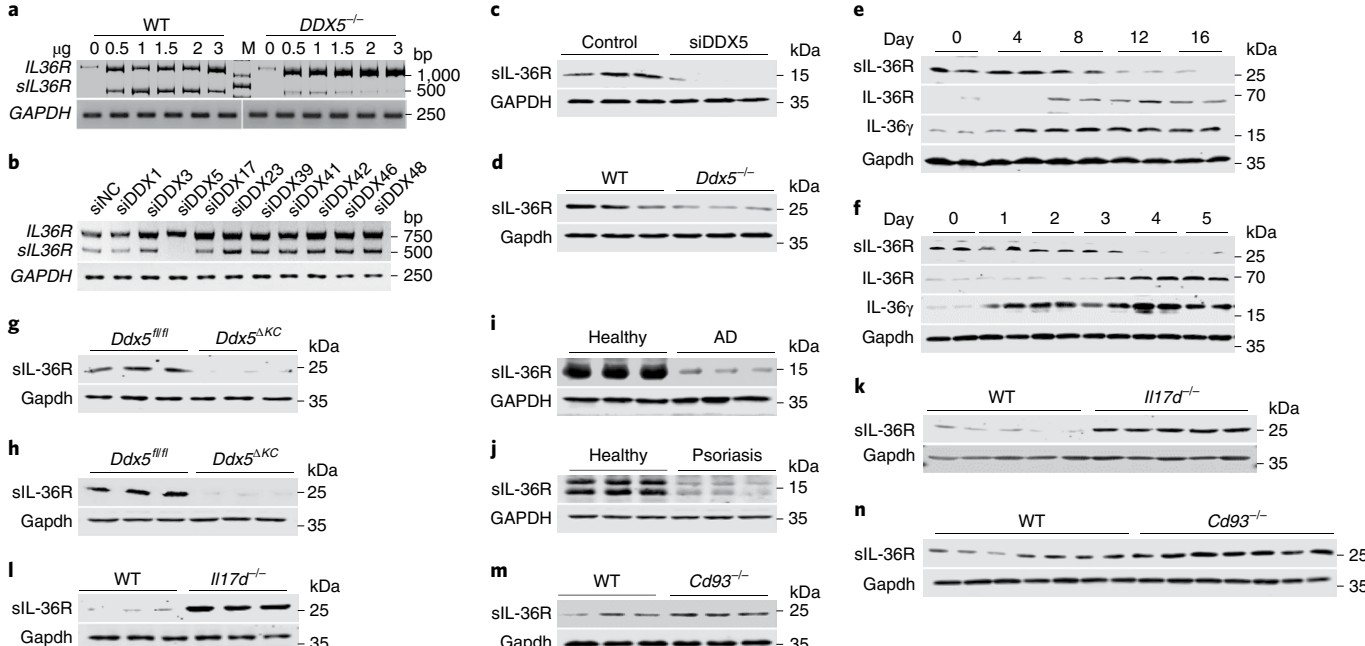

**Fig. 5 | DDX5 regulates *IL36R* pre-mRNA splicing for sIL-36R production. a**, DNA–PAGE of *IL36R* and *sIL36R* transcripts in wild-type and *DDX5*⁻/⁻ HeLa cells transfected with 0–3 µg of human *IL36R*-reporter minigene. **b**, DNA–PAGE of *IL36R* and *sIL36R* transcripts in NHEKs after transduction with siRNA targeting *DDX1*, *DDX3*, *DDX5*, *DDX17*, *DDX23*, *DDX39*, *DDX41*, *DDX42*, *DDX46* and *DDX48*. **c,d**, Immunoblot of sIL-36R in NHEKs transduced with *DDX5*-targeted siRNA (**c**) or murine keratinocytes isolated from *Ddx5*^fl/fl^ and *Ddx5*^ΔKC^ mice (**d**). **e,f**, Immunoblot of sIL-36R, IL-36R and IL-36γ in lesional skin from MC903-treated wild-type mice (**e**) or IMQ-treated wild-type mice (**f**) at the indicated timepoints. **g,h**, Immunoblot of sIL-36R in lesional skin from MC903-treated *Ddx5*^fl/fl^ and *Ddx5*^ΔKC^ mice (**g**) or IMQ-treated *Ddx5*^fl/fl^ and *Ddx5*^ΔKC^ mice (**h**). **i,j**, Immunoblot of sIL-36R in healthy skin and lesional skin from patients with AD (**i**) or psoriasis (**j**). **k,l**, Immunoblot of sIL-36R in lesional skin from MC903-treated wild-type and *Il17d*⁻/⁻ mice (**k**) or IMQ-treated wild-type and *Il17d*⁻/⁻ mice (**l**). **m,n**, Immunoblot of sIL-36R in lesional skin from IMQ-treated wild-type and *Cd93*⁻/⁻ mice (**m**) or MC903-treated wild-type and *Cd93*⁻/⁻ mice (**n**).

compared with IL-36γ-stimulated *DDX5*⁻/⁻ HaCaT cells without *IL1RL2* silencing (Fig. 4c,d).

IL-36γ-stimulated *DDX5*⁻/⁻ HaCaT cells also had increased expression of the *IL36R* mRNA, but not the IL-36R antagonists *IL36RN* or *IL38* (Fig. 4a,e,f and Extended Data Fig. 6h,i). An increase in IL-36R protein was also observed in NHEKs transduced with siRNA targeting *DDX5* and keratinocytes isolated from *Ddx5*^ΔKC^ newborn mice (Fig. 4g,h). Expression of IL-36R protein was increased 4.8-fold or 1.8-fold in whole skin lesions from MC903- or IMQ-treated *Ddx5*^ΔKC^ mice (Fig. 4i and Extended Data Fig. 6j), but reduced by, on average, 68% in whole skin lesions from MC903- or IMQ-treated *Il17d*⁻/⁻ mice or *Cd93*⁻/⁻ mice (Fig. 4j,k and Extended Data Fig. 6k,l), compared with their wild-type counterparts. The expression of IL-36R was increased by 190% or 80% in the skin lesions of patients with AD or psoriasis, respectively, compared with healthy skin (Fig. 4l and Extended Data Fig. 6m). As such, decreased expression of DDX5 amplified the keratinocyte response to IL-36γ by upregulating the expression of IL-36R.

**DDX5 and SF2 regulate pre-mRNA splicing to produce sIL-36R**
Alternative pre-mRNA splicing is a critical step in the posttranscriptional regulation of gene expression[32]. Quantification of splicing events in the RNA-seq datasets from IL-36γ-stimulated wild-type and *DDX5*⁻/⁻ HaCaT cells or skin lesions from MC903- or IMQ-treated *Ddx5*^ΔKC^ and *Ddx5*^fl/fl^ mice indicated that *DDX5* deficiency in keratinocytes changed multiple splicing events (Extended Data Fig. 7a), with a profound effect on exon skipping and the splicing pattern of the genes with skipped exons (Extended Data Fig. 7a,b). To test whether DDX5 regulated the splicing of the *IL1RL2* pre-mRNA, we examined the presence of IL-36R splicing transcripts in primary human and mouse keratinocytes. Two *IL36R* transcripts were amplified by PCR in NHEKs and primary keratinocytes isolated from wild-type newborn mice. Sequencing of

these transcripts identified a human *IL36R* transcript that skipped exon3 and a mouse *Il36R* transcript that skipped exon6 (Extended Data Fig. 7c–f). Both transcripts encoded the ectodomain of IL-36R (Extended Data Fig. 7g,h), a previously unknown soluble sIL-36R.

We next tested whether DDX5 influenced *IL36R* mRNA splicing to generate alternative transcripts. *IL36R* transcripts were increased, whereas *sIL36R* transcripts were decreased in *DDX5*⁻/⁻ HaCaT cells compared with wild-type HaCaT cells, from both the endogenous gene and an *IL36R* reporter minigene transduced in these cells (Fig. 5a,b and Extended Data Fig. 7i). Silencing of the other nine DDX proteins in NHEKs had no effect on the expression of *IL36R* and *sIL36R* (Fig. 5b and Extended Data Fig. 7j). Murine *Il36R* was increased and *sIl36R* was reduced in keratinocytes isolated from *Ddx5*^ΔKC^ newborn mice compared with *Ddx5*^fl/fl^ keratinocytes (Extended Data Fig. 7k). Immunoblotting with an antibody specific for sIL-36R indicated that expression of sIL-36R protein was reduced in *DDX5*-silenced NHEKs or *Ddx5*^ΔKC^ keratinocytes (Fig. 5c,d). In contrast to IL-36R, sIL-36R was gradually reduced in whole skin lesions from wild-type mice at day 8 post-MC903 administration or at day 2 post-IMQ-administration (Fig. 5e,f) and was almost undetectable in skin lesions from patients with AD or psoriasis or from MC903- and IMQ-treated *Ddx5*^ΔKC^ mice (Fig. 5g–j), whereas its expression in skin lesions from MC903- and IMQ-treated *Il17d*⁻/⁻ and *Cd93*⁻/⁻ mice was similar to that in healthy skin from wild-type mice (Fig. 5k–n). These observations indicated that DDX5 deficiency led to *IL36R* pre-mRNA splicing that favored the production of IL-36R.

DDX5 cannot recognize sequence specificity in its RNA substrates[9,33]. Immunoprecipitation and mass spectrometry (MS) indicated that the splicing factor SF2 (also known as SRSF1), which recognizes specific splice sites and defines splicing patterns[34], was the second most abundant protein, after DDX17, that interacted with DDX5 (Extended Data Fig. 8a). The interaction was confirmed by immunoprecipitation

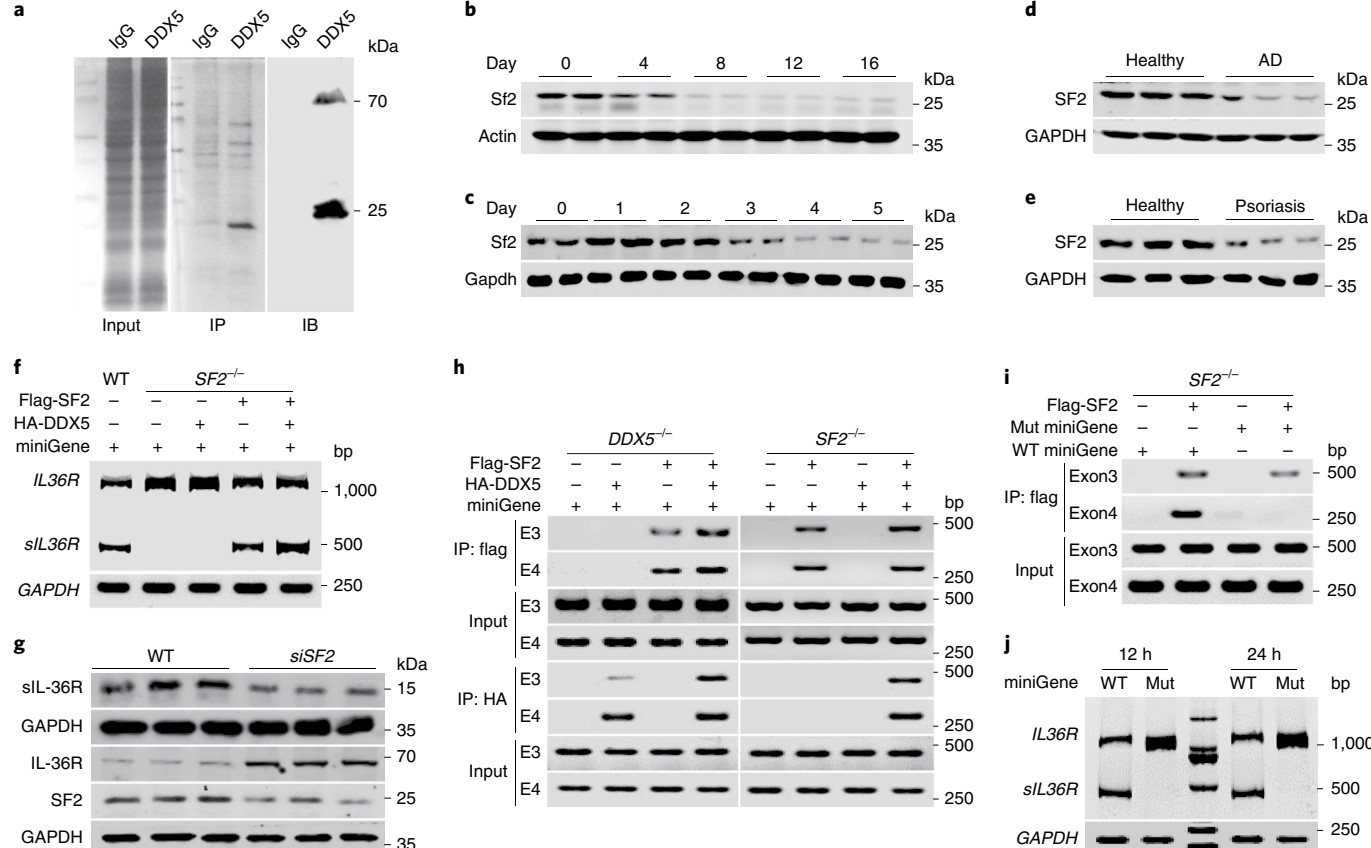

**Fig. 6 | SF2 determines the splicing pattern of *IL36R* pre-mRNA.**
**a**, Immunoprecipitation (IP) of DDX5-binding proteins (middle) by DDX5 antibody in total NHEK cell lysates (left) or DDX5-immunoprecipated proteins incubated with DDX5 and SF2 antibodies for 24 h (right). IB, Immunoblot. **b**,**c**, Immunoblot of Sf2 in the whole skin lesions from MC903-treated (**b**) or IMQ-treated (**c**) wild-type mice at the indicated days. **d**,**e**, Immunoblot of SF2 in healthy skin (*n* = 3) and lesional skin from patients with AD (*n* = 3) (**d**) or psoriasis (*n* = 3) (**e**). **f**, DNA–PAGE analysis of *IL36R* and *sIL36R* transcripts in *SF2*[-/-] HeLa cells transfected with an *IL36R* reporter minigene and exogenous Flag-tagged SF2 and/ or HA-tagged DDX5. **g**, Immunoblot of sIL-36R, IL-36R and SF2 in wild-type NHEKs

and NHEKs transduced with *SF2*-targeted siRNA. **h**, RNA immunoprecipitation of exon3 or exon4 binding to SF2 by anti-Flag beads or to DDX5 by anti-HA beads in the nuclear extracts of *DDX5*[-/-] or *SF2*[-/-] HeLa cells, in which an *IL36R* reporter minigene, exogenous Flag-tagged SF2 and/or HA-tagged DDX5 was expressed, respectively. Horizontal blots 3 and 4 and 7 and 8 are loading controls. **i**, RNA immunoprecipitation of exon4 or exon4 containing a mutation in the ESE binding to SF2. **j**, DNA–PAGE analysis of *IL36R* and *sIL36R* transcripts in HaCaT cells transfected with an *IL36R* reporter minigene or an *IL36R* reporter minigene containing a mutation (Mut) in the ESE in exon4.

---

of SF2 and DDX5 on the same Coomassie Blue-stained gel (Fig. 6a) and by coimmunoprecipitation (Extended Data Fig. 8b,c). Expression of SF2 protein was reduced in the skin lesions of patients with AD and psoriasis or MC903- or IMQ-treated wild-type mice (Fig. 6b–e). Less *sIL36R* transcript from an *IL36R* reporter minigene was detected in *SF2*[-/-] compared with wild-type HeLa cells, and expression was restored by overexpression of SF2, but not DDX5 (Fig. 6f). The siRNA targeting of SF2 in IL-36γ-stimulated NHEKs reduced the expression of sIL-36R, increased the expression of IL-36R (Fig. 6g) and increased the expression of *CCL20*, *CXCL1*, *CXCL6*, *CCL17*, *CCL3*, *CCL11* and *CCL27* compared with IL-36γ-stimulated wild-type NHEKs (Extended Data Fig. 8d). An exonic splicing enhancer (ESE) finder program (http://exon.cshl.edu/ESE)[35] identified multiple putative SF2-binding ESEs in exon3, exon2 and exon4 (Extended Data Fig. 8e) and native RNA immunoprecipitation indicated that SF2 bound predominantly in exon4 of *IL36R* mRNA (Fig. 6h and Extended Data Fig. 8c,e), suggesting that this interaction might mediate exon3 skipping (Extended Data Fig. 8f). SF2 bound to exon3 and exon4 of *IL36R* mRNA in *DDX5*[-/-] HeLa cells, whereas DDX5 did not interact with exon3 and exon4 of *IL36R* mRNA in *SF2*[-/-] HeLa cells (Fig. 6h). Deletion of the ESE in exon4 abrogated SF2 binding (Fig. 6i and Extended Data Fig. 8g) and inhibited the generation of *sIL36R* in HaCaT cells transduced with *IL36R* reporter genes (Fig. 6j).

These data indicated that DDX5 cooperated with SF2 to regulate IL-36R splicing for sIL-36R production in keratinocytes.

## Soluble IL-36R antagonizes IL-36R signaling to control skin inflammation

Next, we determined the role of sIL-36R in skin inflammation. Immunoprecipitation assays showed that sIL-36R bound to IL-36γ and competed with IL-36R for IL-36γ binding in a dose-dependent manner (Fig. 7a), whereas flow cytometry indicated that sIL-36R inhibited the interaction between IL-36R and IL-36γ (Extended Data Fig. 9a). Moreover, overexpression of Flag-tagged sIL-36R decreased the IL-36γ-induced phosphorylation of p65 and p38 MAPK and expression of *CCL20*, *CXCL1*, *CXCL2*, *CCL3*, *CCL17* and *CCL27* in NHEKs (Fig. 7b,c), and *CCL20*, *CXCL1*, *CXCL2*, *CXCL3*, *CCL3*, *CCL11*, *CCL17* and *CCL27* in *DDX5*-silenced NHEKs (Extended Data Fig. 9b), compared with their nontransduced counterparts. Next, we injected recombinant murine sIL-36R into the ears of MC903- or IMQ-treated wild-type mice. Compared with phosphate-buffered saline (PBS) controls, MC903-treated wild-type mice that received sIL-36R had a reduced number of ear skin patches (Extended Data Fig. 9c), decreased ear thickness (Fig. 7d), less expression of *Il4*, *Il13*, *Tslp*, *Ccl11* and *Ccl17* (Fig. 7e) and reduced infiltration of CD45[+] immune cells, including CD11b[+]SiglecF[+] eosinophils, CD49b[+]IgE[+]

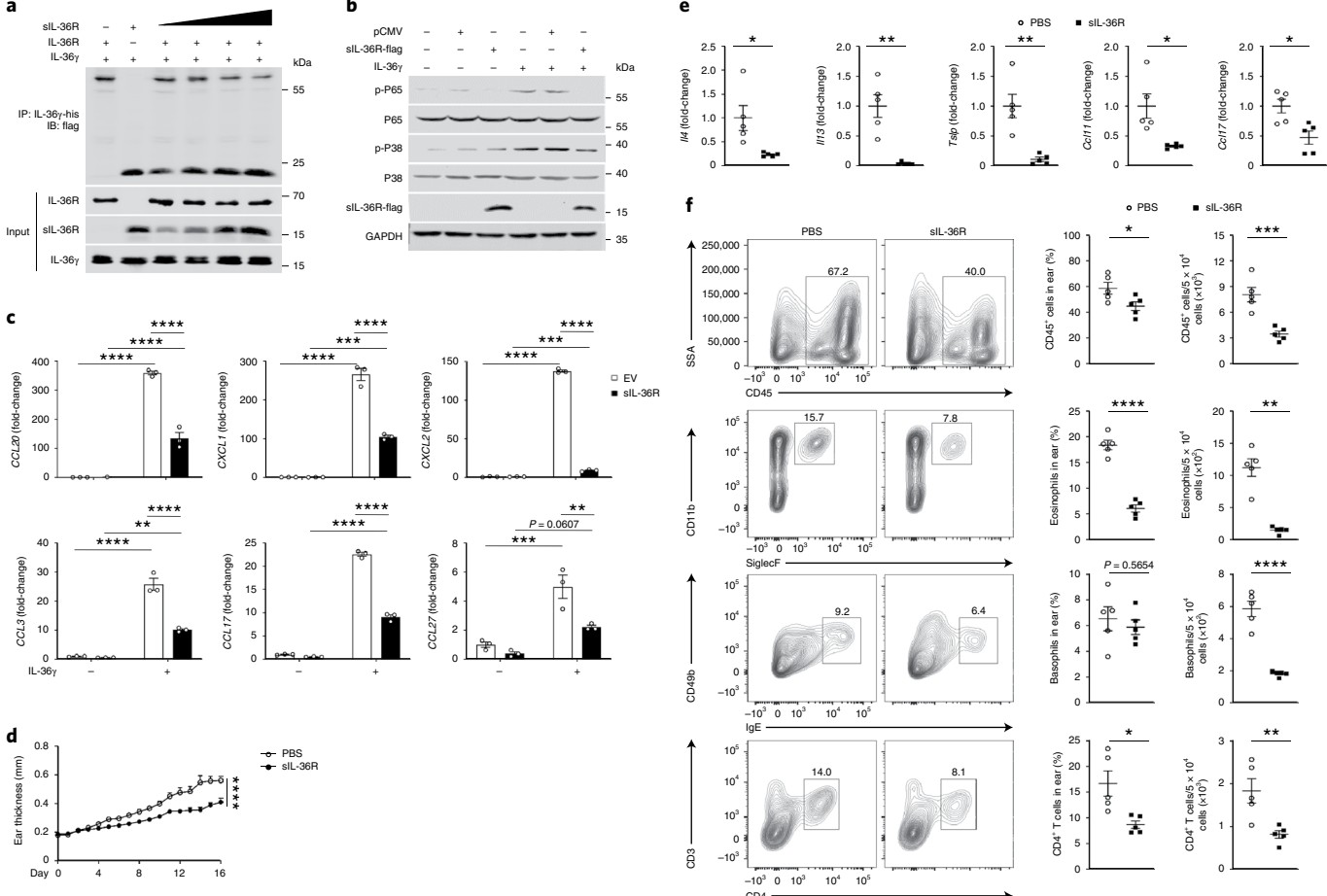

**Fig. 7 | Soluble IL-36R antagonizes IL-36R signaling to inhibit skin inflammation in AD. a**, Immunoprecipitation of IL-36γ-binding IL-36R and sIL-36R in total cell lysates of HeLa cells transduced with plasmids containing Flag-tagged sIL-36R or IL-36R. Total cell lysates were incubated with His-tagged IL-36γ for 4 h. **b**, Immunoblot of p65, p38 MAPK and their phosphorylation in NHEKs treated with IL-36γ with or without overexpression of Flag-tagged sIL-36R. **c**, RT–qPCR of *CCL20*, *CXCL1*, *CXCL2*, *CCL3*, *CCL17* and *CCL27* in NHEKs treated with 100 ng ml⁻¹ of IL-36γ before and after overexpression of Flag-tagged sIL-36R.

EV, empty vector. **d**, Quantification of ear thickness of MC903-treated wild-type mice (*n* = 5) injected intradermally with PBS or 1 μg of recombinant sIL-36R per ear. **e**, RT–qPCR of *Il4*, *Il3*, *Tslp*, *Ccl11* and *Ccl17* in lesional skin from MC903-treated wild-type mice (*n* = 5) treated as in **d**. **f**, Flow cytometry of CD45⁺ cells, CD11b⁺SiglecF⁺ eosinophils, CD49b⁺IgE⁺ basophils and CD3⁺CD4⁺ T cells in ears treated as in **d**. *P < 0.05, **P < 0.01, ***P < 0.001 and ****P < 0.0001. The P values were analyzed by two-way ANOVA (**c** and **d**) or unpaired two-tailed Student's *t*-test (**e** and **f**). Data are presented as mean ± s.e.m.

basophils and CD3⁺CD4⁺ T cells (Fig. 7f). Administration of sIL-36R also reduced ear-skin plaques, decreased epidermal acanthosis, suppressed the production of *Ccl20*, *Cxcl1*, *Il23* and *Il17a* and inhibited the infiltration of CD45⁺ immune cells, including Ly6G⁺ neutrophils, CD45⁺MHCII⁺ APCs and γδTCR⁺ T cells in the skin lesions of IMQ-treated wild-type mice compared with mice that received a PBS injection (Extended Data Fig. 9c–f). These data indicated that sIL-36R antagonized IL-36R signaling to control cutaneous inflammation.

### Soluble IL-36R inhibits cutaneous inflammation in *Ddx5ᐃKC* mice

Next, we investigated whether the decrease in cutaneous sIL-36R was a causative factor for the immunopathology of AD and psoriasis. MC903- or IMQ-treated *Ddx5ᐃKC* mice intradermally injected with sIL-36R had fewer skin patches or plaques (Extended Data Fig. 10a,b), a significant decrease in ear thickness (Fig. 8a,b) and lower expression of IL-4, IL-13, thymic stromal lymphopoietin (TSLP) proteins and *Ccl11*, *Ccl17* and *Ccl22* mRNA in skin lesions from MC903-treated *Ddx5ᐃKC* mice or CCL20, CXCL1, IL-23, IL-17A, IL-17F and TNF proteins in skin lesions from IMQ-treated *Ddx5ᐃKC* mice (Fig. 8c–e) compared with PBS controls. When *Ddx5ᐃKC* mice were crossed with *sIL-36Rᵀᵍ/ᴷᶜ* mice, which express

CAG promoter-driven human sIL-36R in keratinocytes, MC903-treated *Ddx5ᐃKCsIL36Rᵀᵍ/ᴷᶜ* mice did not develop skin plaques, and had thinner ear and epidermis (Fig. 8f and Extended Data Fig. 10c,d) and reduced expression of IL-4, IL-13, TSLP proteins and *Ccl11*, *Ccl17* and *Ccl22* mRNA in lesions (Fig. 8g,h) compared with MC903-treated *Ddx5ᐃKC* mice. In addition, IMQ-treated *Ddx5ᐃKCsIL36Rᵀᵍ/ᴷᶜ* mice did not develop plaques or acanthosis in the skin (Extended Data Fig. 10e,f), and had decreased ear thickness and fewer CCL20, CXCL1, IL-23 and IL-17A proteins (Fig. 8i,j) compared with IMQ-treated *Ddx5ᐃKC* mice. These data indicated that a reduction in cutaneous sIL-36R was a causative factor for the exacerbated inflammation observed in MC903- or IMQ-treated *Ddx5ᐃKC* mice.

## Discussion

In the present study, we observed that IL-17D signaling modulated the expression of DDX5 in keratinocytes, which in turn controlled the extent of inflammation in the skin by regulating the alternative splicing of IL-36R. DDX5 expression in keratinocytes was inhibited by IL-17D through the activation of a CD93–p38 MAPK–AKT–SMAD2/3 signaling pathway and selectively amplified the inflammatory response to IL-36 and aggravated cutaneous inflammation. DDX5 required SF2, which

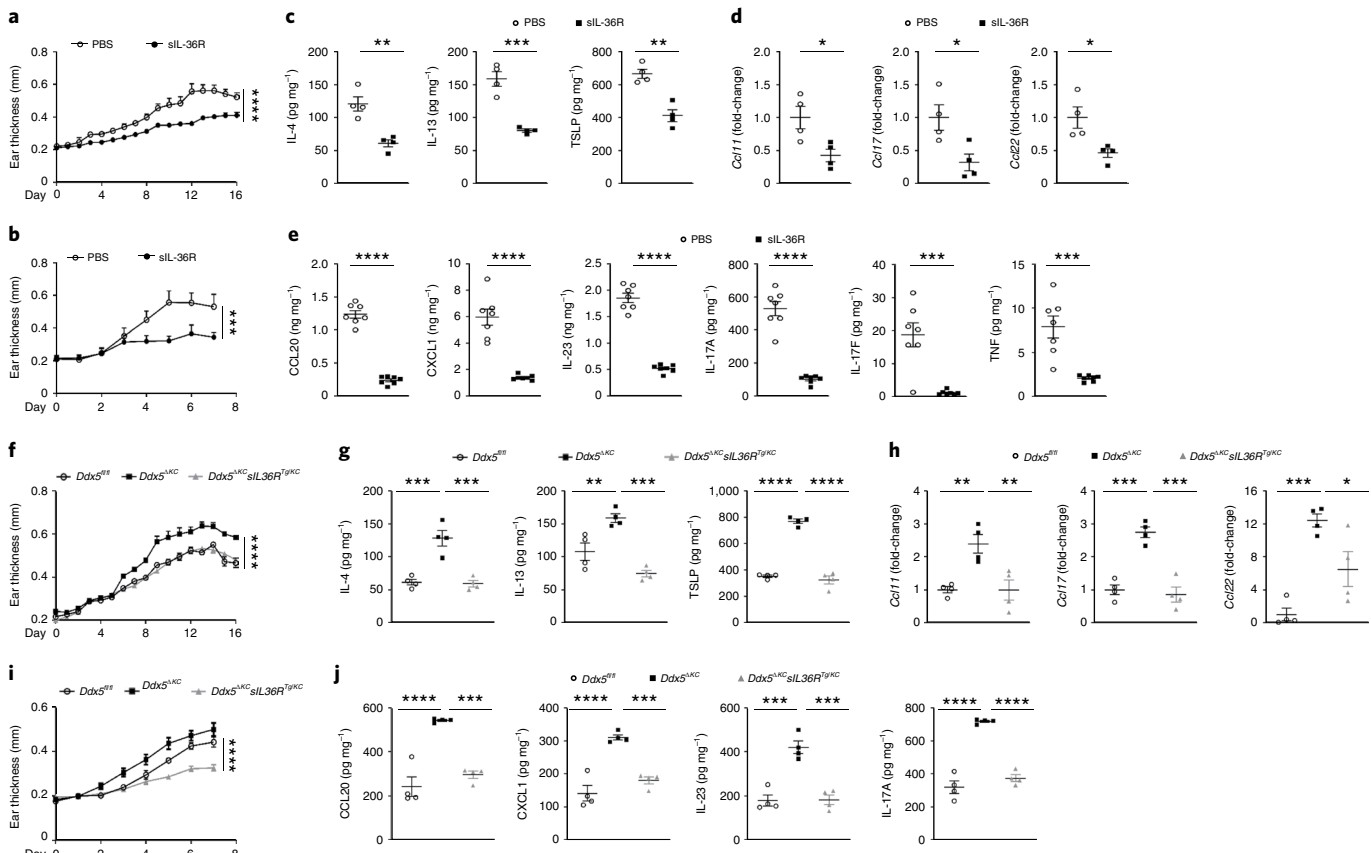

**Fig. 8 | Soluble IL-36R inhibits cutaneous inflammation in *Ddx5^ΔKC* mice. a,b,** Quantification of ear thickness in MC903-treated *Ddx5^ΔKC* mice (*n* = 4) (**a**) or IMQ-treated *Ddx5^ΔKC* mice (*n* = 10) (**b**) intradermally injected with PBS in the left ear or 1 µg of recombinant sIL-36R per ear in the right ear. **c**, ELISA of IL-4, IL-13 and TSLP protein in lesional skin from MC903-treated *Ddx5^Δ/kc* mice injected with sIL-36R as in **a**. **d**, RT–qPCR of *Ccl11*, *Ccl17* and *Ccl22* in lesional skin from MC903-treated *Ddx5^ΔKC* mice injected with sIL-36R as in **a**. **e**, ELISA of CCL20, CXCL1, IL-23, IL-17A, IL-17F and TNF in lesional skin from IMQ-treated *Ddx5^ΔKC* mice injected with sIL-36R as in **b**. **f**, Quantification of ear thickness in MC903-treated *Ddx5^fl/fl* (*n* = 8), *Ddx5^ΔKC* (*n* = 8) and *Ddx5^ΔKCsIL36R^Tg/KC* (*n* = 8) mice. **g**, ELISA of IL-4, IL-13 and TSLP protein in lesional skin from MC903-treated *Ddx5^fl/fl* (*n* = 4), *Ddx5^ΔKC* (*n* = 4) and *Ddx5^ΔKCsIL36R^Tg/KC* (*n* = 4) mice. **h**, RT–qPCR of *Ccl11*, *Ccl17* and *Ccl22* in lesional skin from MC903-treated *Ddx5^fl/fl* (*n* = 4), *Ddx5^ΔKC* (*n* = 4) and *Ddx5^ΔKCsIL36R^Tg/KC* (*n* = 4) mice. **i**, Quantification of ear thickness in IMQ-treated *Ddx5^fl/fl* (*n* = 8), *Ddx5^ΔKC* (*n* = 8) and *Ddx5^ΔKCsIL36R^Tg/KC* (*n* = 8) mice. **j**, ELISA of CCL20, CXCL1, IL-23 and IL-17A in lesional skin from *Ddx5^fl/fl* (*n* = 4), *Ddx5^ΔKC* (*n* = 4) and *Ddx5^ΔKCsIL36R^Tg/KC* (*n* = 4) mice. *P < 0.05, **P < 0.01, ***P < 0.001 and ****P < 0.0001. The *P* values were analyzed by unpaired, two-tailed Student's *t*-test (**c**–**e**) or one-way ANOVA (**g**,**h** and **j**) or two-way ANOVA (**a**,**b**,**f** and **i**). Data are presented as mean ± s.e.m.

recognized ESEs in exon4 of the *IL1RL2* gene, to alternatively splice a soluble form of IL-36R, which antagonized membrane-bound IL-36R to limit cutaneous inflammation.

AD and psoriasis are mediated by distinct T cell polarity and immune responses to T cell-derived cytokines. High-throughput sequencing analyses showed that AD and psoriasis are shaped by dys-regulated immune responses to different cytokines and chemokines when DEGs from AD and psoriatic lesions, compared with nonlesions in their corresponding individuals, were analyzed[6–8]. However, when shared DEGs from nonlesional and lesional skin in patients with AD and psoriasis compared with healthy skin were used for GO and the KEGG pathway enrichment analyses, RNA splicing regulated by shared spli-ceosome complexes was uncovered in both AD and psoriasis. Among all identified shared spliceosomes from GO enrichment analysis, DDX5 is a critical component of multiple spliceosome complexes, such as DDX5/DDX17, DDX5/HNRNPA1 and DDX5/SF2. Our high-throughput RNA-seq results of genetic deletion of *DDX5* in HaCaT cells or keratinocyte *Ddx5* in mice confirmed DDX5 as a key mediator respon-sible for the change of alternative splicing patterns observed in AD and psoriasis

Cytokines such as IL-17A, IL-17E, IL-4 and IL-36γ regulate the inflam-matory responses in keratinocytes in both AD and psoriasis. In addition

to those, we showed that IL-17D signals through CD93–p38 MAPK–AKT–SMAD2/3 to inhibit DDX5 expression in keratinocytes. IL-17D is known to regulate the function of group 3 innate lymphoid cells (ILC3 cells) against intestinal inflammation[23], promote tumor rejection through recruitment of natural killer cells[36] or promote *Listeria* infection by suppressing CD8+ T cell activity[37]. In the present study, we showed that IL-17D regulated IL-36R-mediated skin inflammation through inhibition of DDX5 expression. IL-17D is expressed by endothelial cells, adipo-cytes or epithelial cells, not by T and B cells[23,38]. We did not observe IL-17D expression in skin adipocytes and keratinocytes during AD and psoriasis. Therefore, the cells that express IL-17D in AD and psoriasis remain to be determined. Moreover, CD93 is expressed in multiple cells, including keratinocytes, ILC3 cells, endothelial cells and macrophages. Although we observed that IL-17D regulated keratinocyte inflammation by activation of CD93-mediated signaling, whether IL-17D would regu-late immune responses of other CD93+ cells to drive skin inflammation needs further investigation.

DDX5 acts as a component of the spliceosome and cannot rec-ognize sequence specificity in its RNA substrates[39], suggesting that other cofactors might specifically recognize the alternative splice site in *IL36R* pre-mRNA. Immunoprecipitation and MS identified mul-tiple DDX5-interacting splicing factors, such as serine/arginine-rich

splicing factors (SRSFs) and heterogeneous nuclear ribonucleoproteins (HNRNPs). Among these, SF2 determines splice sites by binding to ESEs in flanking exons of an alternative exon[40,41], whereas HNRNPs block the access of spliceosome elements and inhibit splice-site selection by binding to exonic or intronic splicing silencers[42]. Silencing or deletion of SF2 changed the IL-36R splicing pattern in keratinocytes, whereas silencing of DDX17 or overexpression of HNRNPs did not, excluding their involvement in DDX5-mediated IL-36R splicing. How DDX5 interacts with SF2 to drive IL-36R splicing requires further investigation.

IL-36 cytokines, including IL-36α, IL-36β and IL-36γ, bind and signal through a heterodimeric receptor composed of IL-36R and the IL-1R accessory protein (IL-1RAcP)[43]. IL-36 is elevated in inflamed skin and associates with the pathogenesis of multiple inflammatory skin diseases, such as AD and psoriasis[44,45]. Keratinocyte-released IL-36 cytokines increase IL-4-mediated immunoglobulin (Ig)E production in B cells in AD, and treatment with an IL-36R-blocking antibody decreases IgE production and alleviates the disease phenotype[46]. In psoriasis, IL-36 induces keratinocyte proliferation and dendritic cell activation[47,48]. Deletion of IL-36R in mice reduces the number of dermal IL-17-producing γδ T cells and protects mice from psoriasiform dermatitis[48]. IL-36–IL-36R signaling is suppressed by the IL-36R antagonist (IL-36Ra), which is encoded by *IL36RN* and can bind IL-36R[49]. Missense mutations in *IL36RN* that lead to loss of IL-36Ra expression associate with the development of generalized pustular psoriasis[50]. Our functional and genetic analysis indicated that alternative splicing generated sIL-36R as an additional antagonist of IL-36R that suppressed IL-36R signaling by competing for binding to IL-36 in keratinocytes. Whether IL-36R splicing regulated by the DDX5–SF2 complex and sIL-36R inhibition of IL-36–IL-36R signaling are keratinocyte specific requires further investigation.

In conclusion, our observations indicate that the IL-17D–CD93-mediated decrease in DDX5 expression amplifies cutaneous inflammation and suggest a potential for IL-17D and DDX5 as therapeutic targets in inflammatory skin diseases, whereas the identification of sIL-36R provides insights into the contribution of aberrant RNA splicing to skin inflammation, and may ultimately lead to the development of alternative therapeutic approaches in AD and psoriasis.

## Online content

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

## Methods

### Patients

Skin samples were obtained from seven patients with AD (moderate to severe), ten patients with psoriasis (mild plaque-type psoriasis), three patients with basal cell carcinoma, three patients with squamous cell carcinoma and eight healthy patients with a 2-mm punch biopsy. The information for all the patients is shown in Supplementary Table 1. All these samples were used for protein extraction or paraffin section. Sample acquisitions, including skin biopsies, were approved by the ethics review committees of Huashan Hospital and Shanghai Tenth People's Hospital and performed in accordance with the declaration of Helsinki principles. Informed consent was obtained for all procedures. All patients are volunteers and did not receive compensation.

### Mice

The mice used for OVA induction are BABL/c background and those used for MC903 or IMQ induction are C57BL/6 background. All mice were bred in the specific pathogen-free animal facility at either East China Normal University or Tsinghua University. $DDX5^{fl/fl}$ and $sIL36R^{fl/fl}$ mice were generated by Shanghai Model Organisms Center, Inc. The K14Cre transgenic mice were obtained from Shanghai Model Organisms Center, Inc., whereas the K5Cre transgenic mice were obtained from Xiao Yang lab in the Academy of Military Medical Sciences in China. $DDX5^{fl/fl}$ mice were crossed with the K14Cre transgenic mice to generate $Ddx5^{\Delta KC}$ mice, $sIL36R^{fl/fl}$ mice were crossed with the K5Cre mice to generate $sIL36R^{Tg/KC}$ mice and $Ddx5^{\Delta KC}$ mice were crossed with $sIL36R^{Tg/KC}$ mice to generate $Ddx5^{\Delta KC}sIL36R^{Tg/KC}$ mice. All the animal experiments were performed with the use of the protocols (protocol nos.: m20210233 for AD, m20200316 for psoriasis) approved by the Animal Care and Use Committee at East China Normal University. All surgery was conducted under anesthesia and all efforts were made to minimize suffering. For all animal studies, we performed preliminary experiments to determine the requirements for sample size. Mice were grouped according to genotypes but not performed in a blinded manner. Unless otherwise stated, 8-week-old littermates were used for each animal experiment.

### Cells

Primary murine keratinocytes were isolated from newborn pups (1 or 2 d after birth) and cultured in Medium 154CF (Gibco) supplemented with 0.05 mM $Ca^{2+}$ and human keratinocyte growth supplement (HKGS, Gibco). NHEKs (Lifeline Cell Technology, catalog no. FC-0007) stored in liquid nitrogen were defrosted and cultured in EpiLife medium (Gibco) containing EpiLife Defined Growth Supplement (EDGS, Gibco) and 0.06 mM $Ca^{2+}$ (Gibco). The medium was refreshed every 2 d and cells were subcultured according to the cell fusion. Cells at passages 3–5 were used for subsequent experiments. Immortalized human keratinocyte cell line HaCaT (Cobioer, catalog no. CBP6033) was cultured in RPMI 1640 medium (Gibco) containing 10% fetal bovine serum (FBS; Gibco), 50 U ml$^{-1}$ of penicillin and 50 μg ml$^{-1}$ of streptomycin (Shanghai Yuan Pei) under standard culture conditions. HeLa and HEK293T cells were cultured in Dulbecco's modified Eagle's medium (DMEM; Gibco) containing 10% FBS, 50 U ml$^{-1}$ of penicillin and 50 μg ml$^{-1}$ of streptomycin under standard culture conditions. All cells were cultured at 37 °C and 5% $CO_2$.

### AD mouse model

For the calcipotriol (MC903)-induced AD mouse model, 1 nmol of MC903 (Sigma-Aldrich) dissolved in 10 μl of ethanol was painted on the ears of 8-week-old mice (C57BL/6 background) for 15 d consecutively. Photographs of the ears were taken every day and the ear thickness was measured with vernier calipers (Meinaite) every day. For the OVA-induced AD mouse model, 8-week-old mice were intraperitoneally inoculated with 10 μg of chicken OVA mixed with 4 mg of aluminum hydroxide (ImjectAlum; Thermo Fisher Scientific) in a volume of 200 μl at 1-week intervals (that is, at days 0, 7 and 14). At day 13, the dorsal skin of mice was shaved and tape stripped six times with 3M tape. At day 14, 100 μg of OVA in 100 μl of PBS was placed on a 1-cm$^2$ patch of sterile gauze to make an OVA patch, which was attached on the shaved dorsal skin with a transparent dressing (Tegaderm, 3M) for 7 d (that is, from day 14 to day 20). OVA patches were changed daily. Each mouse had a total of three 1-week exposures to the patch at the same site of separation from each other at 2-week intervals. Dorsal skin or ears were collected for flow cytometry, immunoblotting, RT–qPCR, ELISA or immunofluorescence and hematoxylin and eosin (H&E) staining.

### Psoriasis mouse model

Mice aged 8 weeks (on a C57BL/6 background) were subjected to a daily topical dose of 62.5 mg of IMQ cream (5%) (Shichuan MedShine Pharmaceuticals Co.) on the shaved back for 5 d consecutively or 25 mg per ear for 7 d consecutively. The ear thickness was measured with vernier calipers every day. The dorsal skin or ears were collected for flow cytometry, immunoblotting, RT–qPCR, ELISA or immunofluorescence and H&E staining.

### Intradermal injection of sIL-36R

Each mouse was injected intradermally with 1 μg of purified recombinant murine sIL-36R (dissolved in 20 μl of PBS) in the right ear or 20 μl of PBS in the left ear 1 d before MC903 or IMQ application. On day 16 for MC903-induced AD mice or day 8 for IMQ-induced psoriasis mice, the ears were collected for histology, immunoblotting, ELISA, qPCR or flow cytometric analysis.

### Primary keratinocyte culture and stimulation

Primary murine keratinocytes were isolated from $DDX5^{fl/fl}$ and $DDX5^{\Delta/KC}$ neonates by using 10 mg ml$^{-1}$ of dispase II (Sigma-Aldrich) digestion overnight at 4 °C, followed by 0.05% trypsin–EDTA (Shanghai Yuan Pei) for 10 min at 37 °C. The cells were cultured with Medium 154CF supplemented with 0.05 mM $CaCl_2$ and HKGS. NHEKs were cultured in EpiLife medium containing EDGS and 0.06 mM $Ca^{2+}$. To test keratinocytes in response to cytokines, cells were stimulated by the indicated concentrations of cytokines or inhibitors for the indicated time.

### Gene deletion or silencing

For the depletion of DDX5 and SF2 in HaCaT and HeLa cells, lentiCRIS-PRv2 plasmids containing hSpCas9 and the guide (g)RNA targeting $DDX5$ or $SF2$ in Supplementary Table 2 were constructed. These plasmids were cotransfected HEK293T cells with the packaging plasmids pMD2G (AddGene) and psPAX2 (AddGene) for lentiviral production. HaCaT or HeLa cells were transduced in suspension with 500 μl of viral supernatant in the wells of a 24-well plate. Cells with $DDX5$ or $SF2$ depletion were screened out by using 1 μg ml$^{-1}$ of puromycin. PCR and Immunoblot analyses were used to confirm that DDX5 or SF2 was deleted in these cells.

For $CD93$, $IL-36R$, $DDXs$ or $SF2$ silencing in primary human keratinocytes, siRNAs targeting human $CD93$, $IL-36R$, $DDXs$ and $SF2$ were synthesized by GenePharm. Using lipo3000 transfection reagent (Invitrogen), siRNAs targeting $CD93$, $IL-36R$, $DDXs$ or $SF2$ (Supplementary Table 2) were transfected into primary human keratinocytes at a final concentration of 20 nM. The silencing efficient was analyzed by RT–qPCR.

### RT–qPCR

Mouse ear tissues were homogenized in TRIzol (Takara) using Beadbeater (Biospec), whereas cells were directly resuspended in TRIzol (Takara). Total RNAs were isolated and reverse transcribed into complementary DNA by using RevertAid First Strand cDNA Synthesis Kit (Roche) according to the manufacturer's instructions. RT–qPCR was performed in triplicate using SYBR green master mix (Roche) on a StepOnePlus Real-Time PCR System (Applied Biosystem). Samples

with a low yield of RNA were predetermined and excluded. Results were normalized to glyceraldehyde-3-phosphate dehydrogenase (GAPDH) and the comparative $\Delta\Delta C_T$ method was used to determine the quantification of gene expression. Primers used in this paper are listed in Supplementary Table 3.

### ScRNA-seq data analysis

The processed scRNA-seq data were downloaded from ref. [24] and converted to a Seurat object using SeuratDisk (https://mojaveazure. github.io/seurat-disk). Followed by the typical Seurat workflow (http:// satijalab.org/seurat), 2,000 highly variable genes were normalized, scaled and identified using the NormalizeData, ScaleData and Find-VariableGenes function from Seurat. The number of positive cells was determined by visual inspection of the ElbowPlot. Uniform Manifold Approximation and Projection with a resolution of 0.5 was used to determine cell clusters. The annotation information was reserved for defining clusters. To evaluate the expression of *DDX5* in keratinocytes, fibroblasts and lymphocytes, cell subsets were combined and the expression level was visualized by VlnPlot from Seurat.

### RNA-seq

WT and *DDX5*[−/−] HaCaT cells treated with 100 ng ml[−1] of IL-36γ (Novo-protein) for 4 h or lesional skin from MC903-treated or IMQ-treated *Ddx5*[ΔKC] and *Ddx5*[fl/fl] mice were collected. Total RNAs were isolated with TRIzol. RNA-seq was performed using an Illumina system following Illumina-provided protocols for 2 × 150 paired-end sequencing in WuXi NextCODE at Shanghai, China. To obtain clean reads, FastQC (v.0.11.9) was used to assess the overall quality of raw reads and Trimmomatic (v.0.39)[51] was applied for raw read quality control to cut adapters and remove low-quality reads. Then all the clean reads were mapped on to the human hg38 genome using Hisat2 (v.2.1.0)[52]. Samtools (v.1.7)[53] was used to convert SAM format files into BAM format files and sorted the BAM files. Gene expression levels were quantified by FeatureCounts (v.1.6.3)[54]. Differential expression analysis was performed by R package DESeq2 (v.1.34.0)[55] and the false recovery rate <0.05 was considered to be significantly differentially expressed. GO and KEGG pathway enrichment analyses were performed in ClusterProfiler (v.4.2.0)[56]. Differential alternative splicing events were detected by rMATS (v.3.1.0)[57] and events with $P < 0.05$ were identified as significantly differentially expressed, alternative splicing events.

### ELISA

Skin from MC903- or IMQ-treated mice was homogenized in pre-cooled PBS, pH 7.4, by using Beadbeater (Biospec), and the supernatants of skin homogenate were collected for cytokine evaluation. Cytokine production was measured by ELISA kits of IL-4 (Multisciences (Lianke) Biotech Co., Ltd.), IL-13 (eBioscience), TSLP (Multisciences (Lianke) Biotech Co., Ltd.), IL-23 (R&D), IL-17A (R&D), CCL20 (R&D), CXCL1 (R&D), IL-17F (BD Pharmingen) and TNF (BD Pharmingen) according to the manufacturer's instructions. The samples with a low yield of protein were predetermined and excluded.

### Histology and immunofluorescence staining

Formalin-fixed, paraffin-embedded tissue sections (~5 μm in thickness) mounted on glass slides were used for various methods of staining. The H&E staining was performed as previously described[17]. The epidermal hyperplasia (acanthosis) was evaluated in 12 independent regions of each section. For immunofluorescence, the sections were deparaffinized and pretreated with antigen retrieval solution (10 mM sodium citrate buffer, pH 6.0) for 20 min. The sections were then blocked by 3% bovine serum albumin in PBS for 1 h at room temperature and stained with DDX5 antibody (Abcam) at 4 °C overnight. Next day, the sections were reprobed with rabbit IgG FITC-conjugated antibody (Invitrogen) and then mounted in ProLong Gold antifade reagent with DAPI (Invitrogen) and visualized by confocal microscope (Leica).

### Immunoblotting and immunoprecipitation

The 2-mm pieces of skin taken from patients with AD or psoriasis and MC903-treated, OVA-treated or IMQ-treated mice, or the cells with different treatments, were lysed with pre-cooled radioimmunoprecipitation (RIPA) buffer, pH 7.4, containing protease inhibitor cocktail (Roche). Then, 30 μg of total protein was subjected to sodium dodecyl-sulfate (SDS)–polyacrylamide gel electrophoresis (PAGE) and blotted using the indicated antibodies. For immunoprecipitation, HEK293T or HeLa cells were cultured in DMEM medium and transfected with plasmids containing epitope-tagged DDX5 or SF2. After 24 h, cells were lysed in lysis buffer (250 mM NaCl, 50 mM Hepes, pH 7.4, 1 mM EDTA, 1% Nonidet P-40 (NP-40)) containing protease inhibitor cocktail (Roche). Total protein, 40 μg, was used for immunoprecipitation with anti-Flag beads (Bimake) or anti-hemagglutinin (HA) beads (Bimake) and the precipitated protein complex was used for immunoblotting with antibodies against Flag (MBL) or HA (MBL).

### MS

NHEKs were lysed. Of the cell lysate, 10% was used for input control and 90% was incubated with DDX5 antibody (Abcam) at 4 °C. Next day, protein A/G agarose beads (Beyotime) were added and the incubation was continued for 2–3 h at 4 °C. DDX5-pulldown or rabbit IgG-pulldown proteins were loaded for SDS–PAGE. After Coomassie Brilliant Blue staining, bands with strong intense signals were cut and digested. The resulting peptides were analyzed on the high-pressure liquid chromatography (HPLC) liquid system Dionex Ultimate 3000 (Thermo Fisher Scientific) coupled to a Dionex Trap column (100 μm × 2 cm × 5 μm) with an in-house packed C18 column (75 μm × 15 cm × 3 μm), and the mass of peptides was analyzed by the maXis HD-UHR-TOF mass spectrometer (Bruker). The spectra from MS were automatically used for searching against the nonredundant International Protein Index human protein database (v.3.72) with the Bioworks browser (rev.3.1).

### Skin cell preparation and flow cytometry

Skin cells were prepared according to previous studies with minor modifications[58]. In general, the epidermis and dermis were separated using dispase II (5 mg ml[−1] in Hanks' Balanced Salt Solution; 37 °C for 90 min). The dermal cells were separated by collagenase (Roche) and hyaluronidase (Sigma-Aldrich) digestion (10 mM Hepes, collagenase D (2.5 mg ml[−1]), hyaluronidase (100 U ml[−1]) and DNase (50 μg ml[−1]) in DMEM; 37 °C for 45 min). Isolated cells were stained with different cell surface markers (CD45 for leukocytes; CD45[+] and IgE[+] for basophils; CD45[+] and SiglecF[+] for eosinophils; CD45[+], CD3[+] and CD4[+] for CD4[+] T cells; CD45[+], CD11b[+] and Ly6G[+] for neutrophils; CD45[+], CD11c[+] and MHCII[+] for APCs; and CD45[+], CD3[+] and γδT cell receptor (TCR[+]) for γδT cells). The cells were then fixed and the relevant isotype control monoclonal antibodies were used. Samples were analyzed using LSR Fortessa (BD Biosciences) and FlowJo v.10 software (TreeStar).

### Transfection of *IL36R* splicing reporter

*IL36R*-reporter minigene consisted of the genomic region of *IL36R* encompassing 139 bp of intron2, exon3, 201 bp of intron3 and exon4 was cloned in the pcDNA3.1 plasmid. PCR primers for *IL36R* reporter minigene constructs are listed in Supplementary Table 3. Two versions of the *IL36R*-reporter minigene containing the wild-type exon4 or mutation of the ESE in the context of exon4 were constructed and transfected into HeLa cells using Lipofectamine 2000 (Invitrogen) according to the manufacturer's recommendations. Total RNA was harvested 24 h after transfection using TRIzol reagent.

### Characterization of *IL36R* splicing

For *IL36R* splicing analysis, total RNA was extracted and reverse transcription was performed using RevertAid First Strand cDNA Synthesis Kit (Roche) according to the manufacturer's instructions. PCR reactions (20 μl) were prepared as follows: 2X Taq Master (Novoprotein), 10 μl;

*IL36R* Exon3 Primer (forward), 0.5 µl; *IL36R* Exon4 Primer (reverse), 0.5 µl; cDNA, 2 µl; and ddH$_2$O, 7 µl. PCR primers for endogenous *IL36R* were complementary to exon2 (forward) and exon5 (reverse) (Supplementary Table 3). PCR products were electrophoresed on 12% nondenaturing polyacrylamide/TBE gels.

## Native RIP
RNA immunoprecipitation (RIP) was performed as previously described[59] with minor modification. WT, *DDX5$^{-/-}$* and *SF2$^{-/-}$* HeLa cells were transfected with an *IL36R*-reporter minigene and plasmids containing HA-tagged DDX5 or Flag-tagged SF2, respectively. After 24 h, cells were collected and suspended in an equal volume of polysome lysis buffer (100 mM KCl, 5 mM MgCl$_2$, 10 mM Hepes, pH 7.0, 0.5% NP-40, 1 mM dithiothreitol (DTT), 100 units ml$^{-1}$ of RNase and 400 µM RVC (New England Biolabs)) supplemented with protease inhibitors. Cell lysates were centrifuged at 15,000*g* for 15 min to remove large particles. Before incubation with cell lysates, HA or Flag beads were washed by 1 ml of ice-cold NT2 buffer (50 mM Tris-HCl, pH 7.4, 150 mM NaCl, 1 mM MgCl$_2$ and 0.05% NP-40) 4× and then resuspended in 850 µl of the immunoprecipitation reaction solution (200 units of an RNase inhibitor, 400 µM RVC, 10 µl of 100 mM DTT, 30 µl of 0.5 mM EDTA and 800 µl of cold NT2 buffer). Cleared cell lysates (20 mg ml$^{-1}$), 150 µl, were incubated with 850 µl of HA/Flag beads solution at 4 °C for 4 h and then centrifuged at 366*g* for 2 min to pellet beads. Beads were then washed with 1 ml of ice-cold NT2 buffer 4–5× and TRIzol was added to isolate total RNA from messenger ribonucleoprotein components binding on beads. Total RNA was reverse transcribed into cDNA by using PrimeScript RT Reagent Kit with gDNA Eraser (Takara) according to the manufacturer's instructions and specific primers for exon3 and exon4 were used to amplify SF2- or DDX5-binding fragments.

## Competition binding assay
HeLa cells were seeded into a 6-well plate and cultured in DMEM. The cells were grown to 70% confluence and then transfected with the indicated doses of plasmids containing Flag-tagged *IL36R* gene or Flag-tagged *sIL36R* gene, or transfected with plasmids containing green fluorescent protein (GFP)–*IL36R* gene or *sIL36R* gene. After 36 h, cells transfected with plasmids containing GFP–IL-36R or sIL-36R were stimulated by 100 ng ml$^{-1}$ of His-tagged IL-36γ. After 30 min, cells were collected and stained with phycoerythrin–anti-His antibody for 30 min, followed by flow cytometric analysis. For competition binding analyzed by immunoprecipitation, cells transfected with plasmids containing Flag-tagged *IL36R* gene or Flag-tagged *sIL36R* gene were lysed with radioimmunoprecipitation assay buffer, and 1 µg of His-tagged IL-36γ was added. After 4 h, IL-36γ-binding IL-36R or sIL-36R was pulled down by anti-His beads (Bimake) and loaded on to SDS–PAGE for immunoblotting with anti-Flag antibody.

## Statistics and reproducibility
In vitro experiments: experiments were done in triplicate and independently repeated three times with similar results, with few exceptions in which experiments were repeated twice. For each experiment every sample was processed identically and internal controls and normalization methods were included to avoid technical bias. In vivo experiments: experiments were independently repeated twice with similar results. An exact *n* for each experimental group/condition is shown in the figures by symbols and in Supplementary Table 4. Each symbol represents an individual mouse. All data are presented as mean ± s.e.m. We used unpaired, two-tailed Student's *t*-test to determine significance between two groups. We did analyses of multiple groups by one-way or two-way analysis of variance (ANOVA) with Bonferroni's posttest of GraphPad Prism v.9. For all statistical tests, we considered *P* < 0.05 to be statistically significant. *P* values of RT–qPCR, ELISA and flow cytometric analyses are reported in Supplementary Table 4.

No statistical methods were used to predetermine sample sizes, but our sample sizes are similar to those reported in previous publications[15,17,23]. Data distribution was assumed to be normal but this was not formally tested.

## Reporting summary
Further information on research design is available in the Nature Research Reporting Summary linked to this article.

## Data availability
The raw sequence data reported in the present paper have been deposited in the GEO database under accession nos. GSE208666, GSE208669 and GSE208671. The MS proteomics data have been deposited in the ProteomeXchange Consortium (http://www.proteomexchange.org)[60] under accession no. PXD021379. All other data supporting the findings of the present study are available within the paper or from the corresponding author upon request. Source data are provided with this paper.

## Code availability
The raw sequence code reported in the present paper has been deposited in the GEO database under accession nos. GSE208666, GSE208669 and GSE208671. The MS proteomics code has been deposited in the ProteomeXchange Consortium (http://www.proteomexchange.org)[60] under accession no. PXD021379. All other code supporting the findings of thhis present study are available within the paper or from the corresponding author upon request.

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

## Acknowledgements
This work is supported by the National Natural Science Foundation of China (grant nos. 82071785 and 31670925 to Y.L.), the National Key Research and Development Program of China (grant no. 2016YFC0906200/2016YFC0906202 to Y.L.), the ECNU Multifunctional Platform for Innovation (011) and the Innovation Program of Shanghai Municipal Education Commission (grant no.

2019-01-07-00-07-E00046 to Y.S.). We thank C. Xu from Shanghai Institute of Biochemistry and Cell Biology, Chinese Academy of Sciences and Y. Zhang from University of California, San Diego for critical reading and helpful suggestions.

## Author contributions

Y.L. conceived the project. Y.L., X.N., Y.X. and W.W. designed the experiments. X.N., Y.X., W.W., B.K., M.Y., Y.W., Q.C., X.W., H.L., X.G., H.G., L.C. and Z.C. performed the experiments. X.N., Y.X., W.W., J.O., J.C., R.Z., T.S., R.Z. and Y.L. analyzed the data. L.C., Z.C., Y.S. and W.L. collected human samples. J.H. and C.D. provided *Il17d*[superscript −/−] mice, *Cd93*[superscript −/−] mice and CD93 antibody. W.L., L.-F.W. and C.D. provided intellectual advice and helped with manuscript editing. Y.L. wrote the manuscript with input from all authors.

## Competing interests

Y.L., X.N., Y.X., W.W., Y.W., B.K. and X.G. have filed provisional patents (202110117195.8 and 201910333393.0) disclosure based on the results in the paper. The remaining authors declare no competing interests.

## Additional information

**Extended data** is available for this paper at https://doi.org/10.1038/s41590-022-01339-3.

**Correspondence and requests for materials** should be addressed to Yuping Lai.

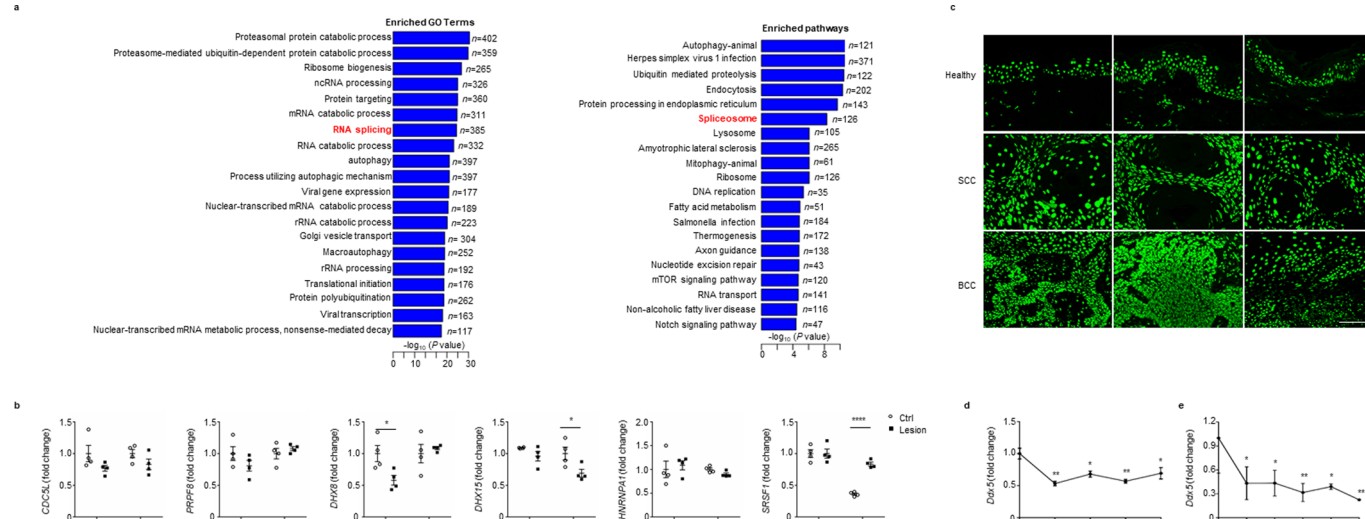

**Extended Data Fig. 1 | DDX5 in AD, psoriasis and skin carcinomas. a**, GO and KEGG pathway analyses of 13,934 shared DEGs in nonlesional skin and lesional skin from AD and psoriasis in RNA-seq datasets from GEO database (GSE121212). **b**, RT-qPCR of *CDC5L*, *PRPF18*, *DHX8*, *DHX15*, *HNRNPA1* and *SRSF1* in lesional skin from AD and psoriasis. **c**, Immunofluorescence analysis of DDX5+ cells in healthy skin or skin from patients with squamous cell carcinoma (SCC) or basal cell carcinoma (BCC). Scale bar, 50μm. **d-e**, RT-qPCR of *Ddx5* in lesional skin from MC903-treated (Day 0 $n$ = 4, Day 4,8,12,16, $n$ = 3) (**d**) or IMQ-treated ($n$ = 3) (**e**) wild-type mice at indicated days. Data represent two independent experiments. *$P$ < 0.05, **$P$ < 0.01 and ****$P$ < 0.0001. $P$ values were analyzed by two-way ANOVA (**c**) or one-way ANOVA (**d-e**). Data are presented as mean ± s.e.m.

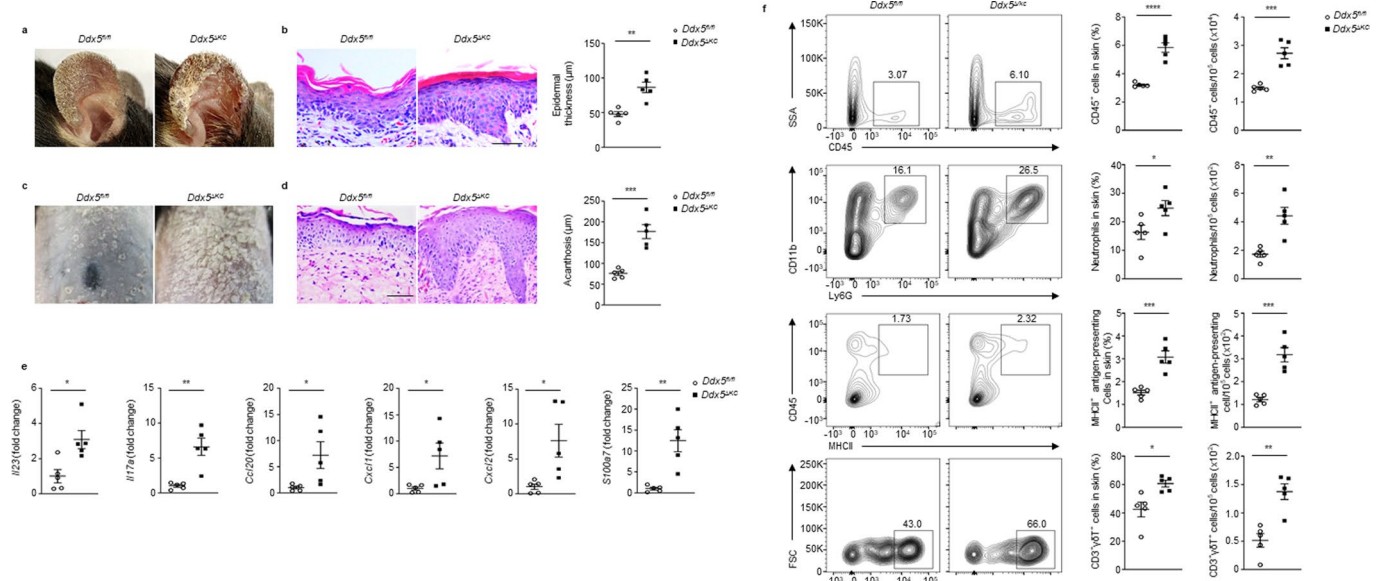

**Extended Data Fig. 2 | DDX5 deficiency in keratinocytes drives skin inflammation. a**, Representative photos of ears from $Ddx5^{fl/fl}$ ($n = 5$) and $Ddx5^{ΔKC}$ mice ($n = 5$) 16 days post-MC903 administration. **b**, H&E analysis of skin sections from $Ddx5^{fl/fl}$ and $Ddx5^{ΔKC}$ mice treated as in (**a**) and quantification of epidermal thickness of skin sections. Scale bar, 50μm. **c**, Representative photos of dorsal skin from $Ddx5^{fl/fl}$ and $Ddx5^{ΔKC}$ mice 5 days post-IMQ administration. **d**, H&E analysis of skin sections from $Ddx5^{fl/fl}$ and $Ddx5^{ΔKC}$ mice treated as in (**c**) and quantification of acanthosis of skin sections. Scale bar, 50μm. **e**, RT-qPCR of $Il23$, $Il17a$, $Ccl20$, $Cxcl1$, $Cxcl2$ and $S100a7$ in lesional skin treated as in (**c**). **f**, Flow cytometry of CD45+ cells, CD11b+Ly6G+ neutrophils, MHCII+ APCs and CD3+γδTCR+ cells in lesional dorsal skin from IMQ-treated $Ddx5^{fl/fl}$ ($n = 5$) and $Ddx5^{ΔKC}$ mice ($n = 5$). Data represent two independent experiments. *$P < 0.05$, ** $P < 0.01$, *** $P < 0.001$ and **** $P < 0.0001$. $P$ values were evaluated by unpaired, two-tailed Student's $t$-test. Data are presented as mean ± s.e.m.

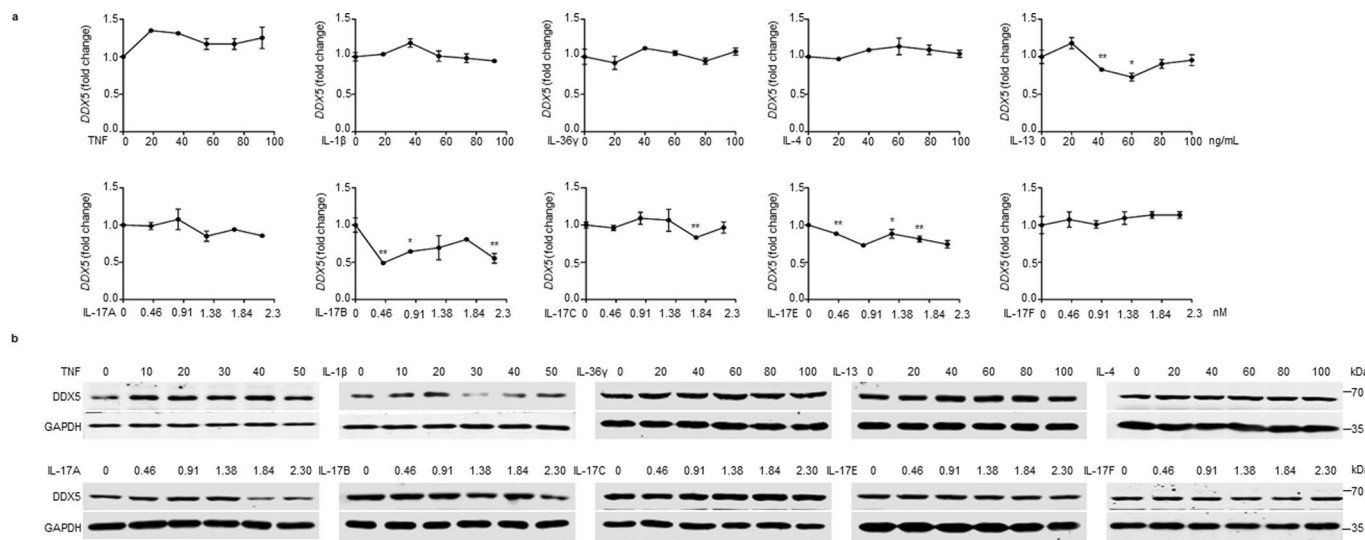

**Extended Data Fig. 3 | DDX5 expression in keratinocytes regulated by different cytokines. a**, RT-qPCR of *DDX5* in keratinocytes (*n* = 3) treated with different doses of TNF, IL-1β, IL-36γ, IL-4, IL-13, IL-17A, IL-17B, IL-17C, IL-17E and IL-17F. Data represent three independent experiments. **b**, Immunoblot of DDX5 in keratinocytes treated with different doses of TNF, IL-1β, IL-36γ, IL-4, IL-13, IL-17A, IL-17B, IL-17C, IL-17E and IL-17F. Data represent three independent experiments. *$P$ < 0.05 and **$P$ < 0.01. $P$ values were analyzed by one-way ANOVA. Data are presented as mean ± s.e.m.

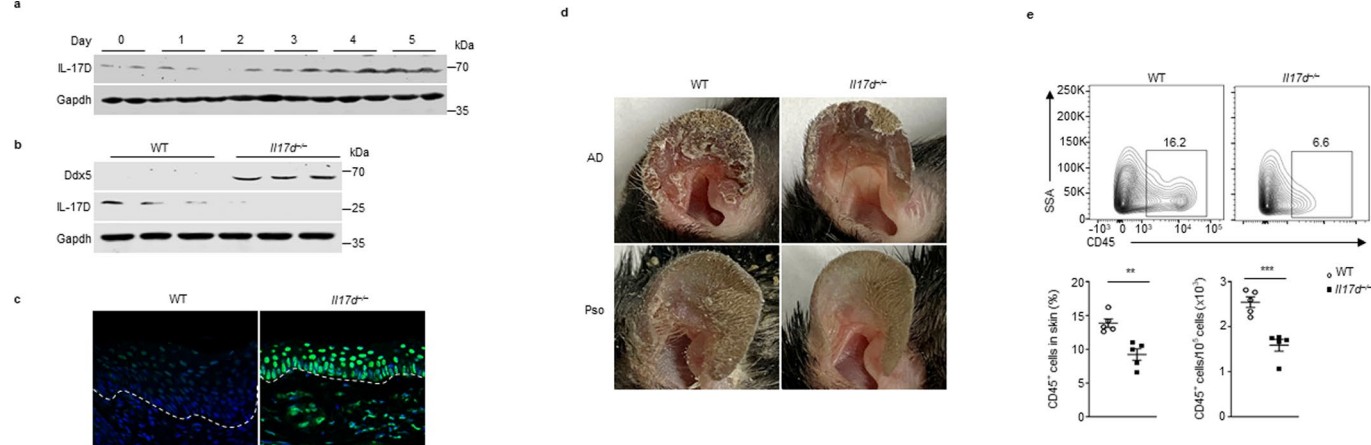

**Extended Data Fig. 4 | IL-17D inhibits DDX5 to drive skin inflammation. a**, Immunoblot of Ddx5 in lesional skin from IMQ-treated wild-type mice ($n = 2$) at day 0 to day 5. **b**, Immunoblot of Ddx5 and IL-17D in lesional skin from IMQ-treated wild-type ($n = 3$) and *Il17d*$^{-/-}$ mice ($n = 3$). **c**, Immunofluorescence analysis of Ddx5$^+$ cells in lesional skin from IMQ-treated wild-type and *Il17d*$^{-/-}$ mice ($n = 5$). Scale bars, 25μm. The dotted lines indicate the edge between epidermis and dermis. **d**, Representative photos of ears from MC903- or IMQ-treated wild-type ($n = 5$) and *Il17d*$^{-/-}$ mice ($n = 5$). **e**, Flow cytometry of CD45$^+$ cells in lesional skin from IMQ-treated wild-type ($n = 5$) and *Il17d*$^{-/-}$ mice ($n = 5$). Data represent two independent experiments. **$P < 0.01$ and ***$P < 0.001$. *P* values were analyzed by unpaired, two-tailed Student's *t*-test. Data are presented as mean ± s.e.m.

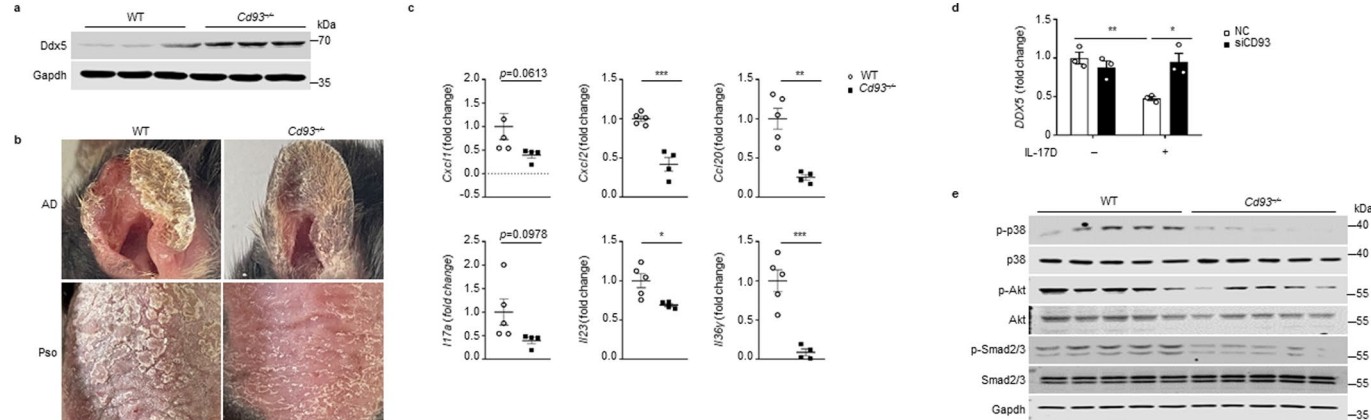

**Extended Data Fig. 5 | Activation of CD93 inhibits DDX5 to drive skin inflammation. a**, Immunoblot of Ddx5 in lesional skin from IMQ-treated wild-type and *Cd93*[-/-] mice (*n* = 3). **b**, Representative photos of ears from MC903-treated (*n* = 7) or IMQ-treated (*n* = 4) wild-type and *Cd93*[-/-] mice. **c**, RT-qPCR of *Cxcl1*, *Cxcl2*, *Ccl20*, *Il17a*, *Il23* and *Il36γ* in lesional skin from IMQ-treated wild-type (*n* = 5) and *Cd93*[-/-] mice (*n* = 4). **d**, RT-qPCR of *DDX5* in NHEKs (*n* = 3) treated with

2.3 nM IL-17D after CD93 silencing. **e**, Immunoblot of p38, Akt, Smad2/3 and their phosphorylation in lesional skin from IMQ-treated wild-type (*n* = 5) and *Cd93*[-/-] mice (*n* = 5). Data represent two independent experiments. *$P < 0.05$, ** $P < 0.01$ and *** $P < 0.001$. *P* values were determined by unpaired, two-tailed Student's *t*-test (**e**) or two-way ANOVA (**d**). Data are presented as mean ± s.e.m.

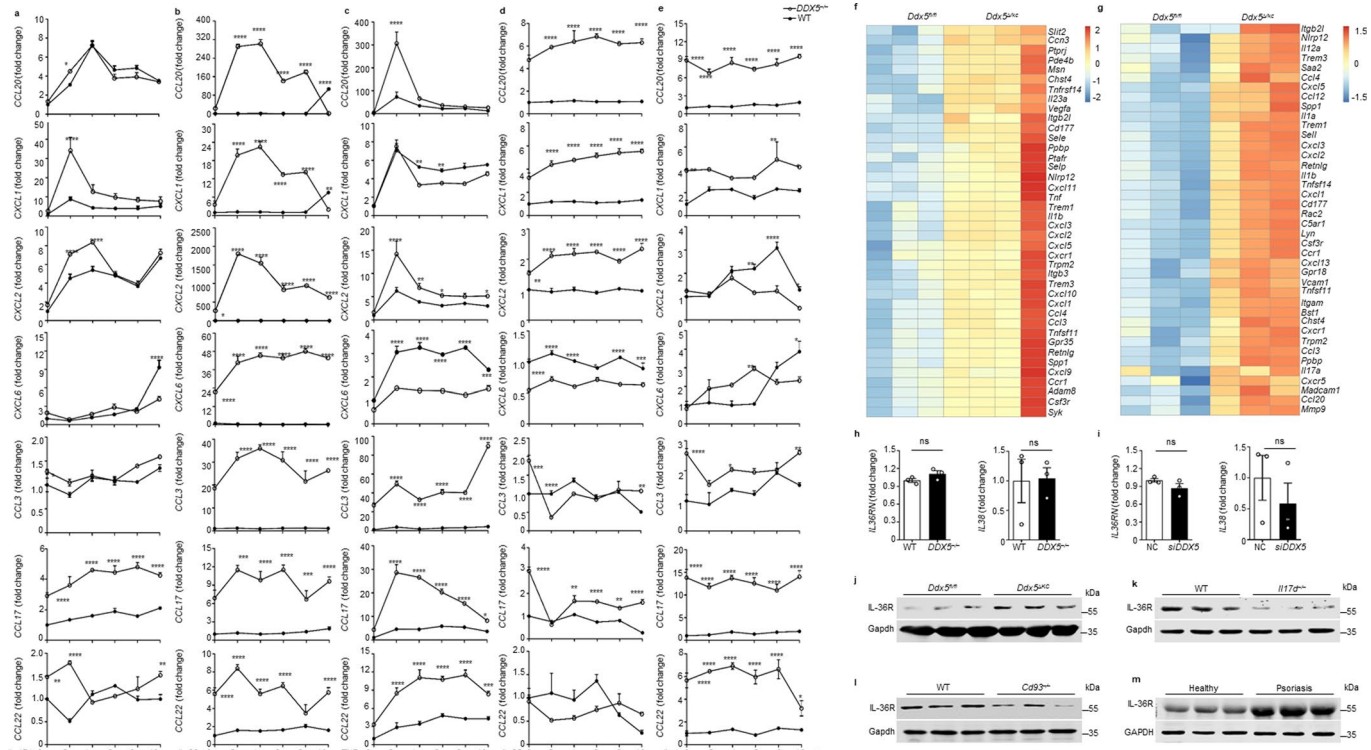

**Extended Data Fig. 6 | DDX5 mediates immune responses in keratinocytes and psoriasis. a-e**, RT-qPCR of *CCL20*, *CXCL1*, *CXCL2*, *CXCL6*, *CCL3*, *CCL17* and *CCL22* in WT and *DDX5^−/−* HaCaT cells (*n* = 3) treated with IL-17A (**a**), IL-36γ (**b**), TNF (**c**), IL-25 (**d**) or IL-4 (**e**) for 0-10 hours. Data represent three independent experiments. **f-g**, Heatmap of top 40 up-regulated genes related to leukocyte migration in MC903-treated *Ddx5^fl/fl^* (*n* = 3) and *Ddx5^ΔKC^* mice (*n* = 4) (**f**) or IMQ-treated *Ddx5^fl/fl^* (*n* = 3) and *Ddx5^ΔKC^* mice (*n* = 3) (**g**) by RNA-seq analysis. **h-i**, RT-qPCR of *IL36RN* and *IL38* in WT and *DDX5^−/−* HaCaT cells (*n* = 3) (**h**) or in NHEKs

(*n* = 3) with DDX5 silencing (**i**). Data represent three independent experiments. **j-l**, Immunoblot of IL-36R in lesional skin from IMQ-treated *Ddx5^fl/fl^* and *Ddx5^ΔKC^* mice (*n* = 3) (**j**) or IMQ-treated wild-type and *Il17d^−/−* mice (*n* = 3) (**k**) or IMQ-treated wild-type and *Cd93^−/−* mice (*n* = 3) (**l**). **m**, Immunoblot of IL-36R in healthy skin and lesional skin from patients with psoriasis (*n* = 3). Data represent two independent experiments. *P < 0.05, **P < 0.01, ***P < 0.001 and ****P < 0.0001. n.s. no significance. *P* values were analyzed by two-way ANOVA (**a-e**) or unpaired, two-tailed Student's *t*-test (**f,g**). Data are presented as mean ± s.e.m.

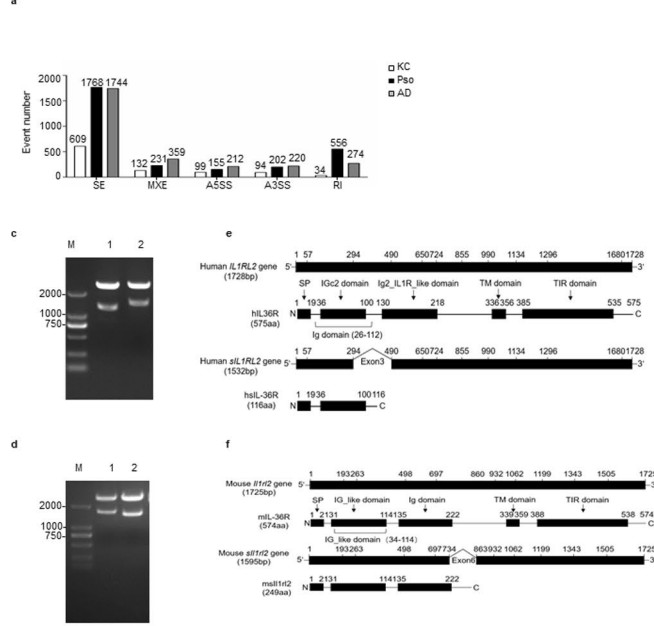

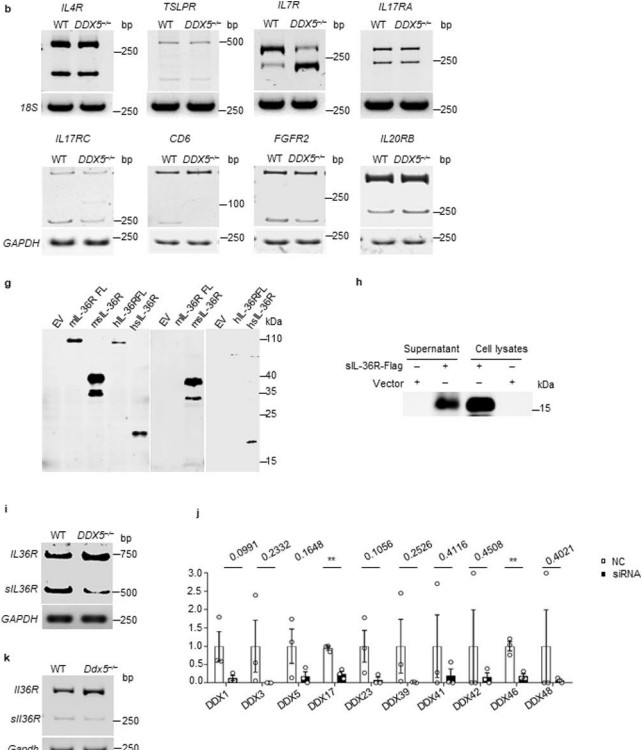

**Extended Data Fig. 7 | DDX5 mediates the change of splicing events and IL-36R/sIL-36R expression. a**, Analysis of different splicing events in IL-36γ-stimulated wild-type and *DDX5*⁻/⁻ HaCaT cells (*n* = 3), MC903-treated *Ddx5*^fl/fl^ (*n* = 3) and *Ddx5*^ΔKC^ (*n* = 4) mice or IMQ-treated *Ddx5*^fl/fl^ and *Ddx5*^ΔKC^ mice (*n* = 3). SE, skipped exon; MXE, mutually exclusive exon; A5SS, alternative 5' splice site; A3SS, alternative 3' splice site; RI, retained intron. **b**, DNA-PAGE of splicing variants of *IL4R*, *TSLPR*, *IL7R*, *IL17RA*, *IL17RC*, *CD6*, *FGFR2* and *IL20RB* in wild-type and *DDX5*⁻/⁻ HaCaT cells. Data represent three independent experiments. **c-d**, Agarose gel analysis of *IL36R* transcripts amplified by PCR in NHEKs (**c**) or primary murine keratinocytes (**d**). **e-f**, Schematic diagrams of the structure of human (**e**) or murine (**f**) IL-36R and sIL-36R. **g**, Immunoblot of full-length IL-36R or sIL-36R by the antibody against Flag (left panel) or the specific antibody against murine sIL-36R (Middle panel) or the specific antibody against human sIL-36R (Right panel). **h**, Immunoblot of sIL-36R in cell cultures or cell lysates from HaCaT cells in which Flag-tagged human sIL-36R was overexpressed. Data represent three independent experiments. **i**, DNA-PAGE of *IL36R* and *sIL36R* transcripts in wild-type and *DDX5*⁻/⁻ HaCaT cells. **j**, Silencing efficiency of indicated genes in NHEKs (*n* = 3). **k**, DNA-PAGE of *Il36R* and *sIl36R* transcripts in primary murine keratinocytes isolated from *Ddx5*^fl/fl^ and *Ddx5*^ΔKC^ newborn mice. Data represent three independent experiments. *P < 0.05, **P < 0.01, ***P < 0.001 and ****P < 0.0001. n.s. no significance. *P* values were analyzed by two-way ANOVA (**j**). Data are presented as mean ± s.e.m.

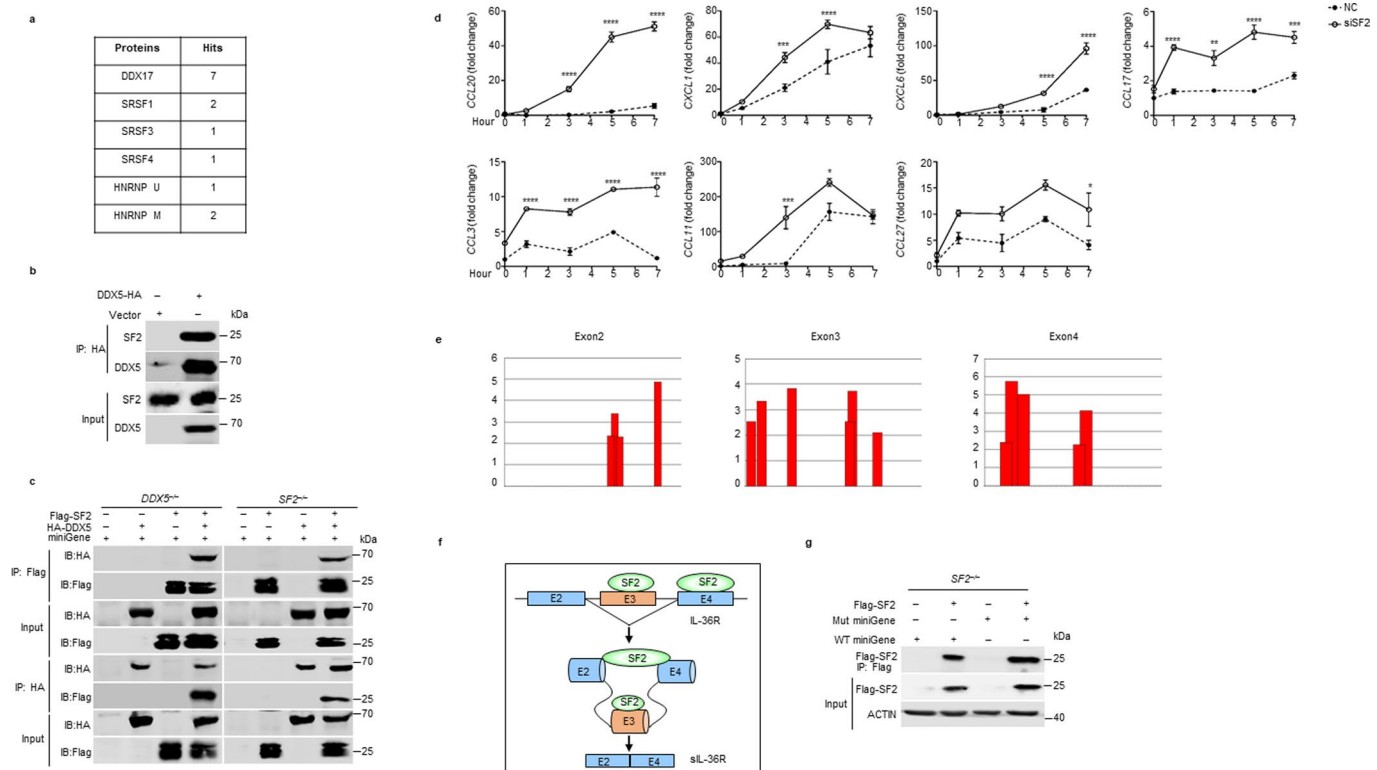

**Extended Data Fig. 8 | SF2 binds to DDX5 and regulates IL-36R splicing and inflammatory responses in keratinocytes. a**, Mass spectrum analysis of DDX5-binding splicing factors in NHEKs. **b**, Co-immunoprecipitation analysis of the interaction of DDX5 and SF2. **c**, Immunoblot of HA-tagged or Flag-tagged proteins that bind to *IL36R* pre-mRNA by Flag-tagged antibody or HA-tagged antibody. **d**, RT-qPCR of *CCL20*, *CXCL1*, *CXCL6*, *CCL17*, *CCL3*, *CCL11* and *CCL27* in NHEKs in response to 100 ng/mL IL-36γ before and after *SF2* was silenced. **e**, SF2-binding ESEs on exon 2, exon 3 and exon 4 of human *IL1RL2* gene analyzed by the ESEfinder program (http://exon.cshl.edu/ESE). **f**, Schematic diagram represents *IL36R* pre-mRNA splicing regulated by SF2. Strong interaction of SF2

on the flanking exon 4 is responsible for skipping of the alternative exon 3 to generate sIL-36R. When the interaction of SF2 on exon 4 is weakened, either by ESE mutation in exon 4 or SF2 depletion, the alternative exon 3 is selected for IL-36R generation. **g**, Immunoblot of *IL36R* pre-mRNA-binding SF2 in nuclear extracts prepared from *SF2⁻/⁻* HeLa cells by Flag-tag antibody, in which *IL36R* reporter minigene or *IL36R* reporter minigene containing ESE mutation in Exon 4 was co-transfected with exogenous Flag-tagged SF2. Data represent at least three independent experiments. *$P < 0.05$, **$P < 0.01$, ***$P < 0.001$ and ****$P < 0.0001$. $P$ values were determined by two-way ANOVA. Data are presented as mean ± s.e.m.

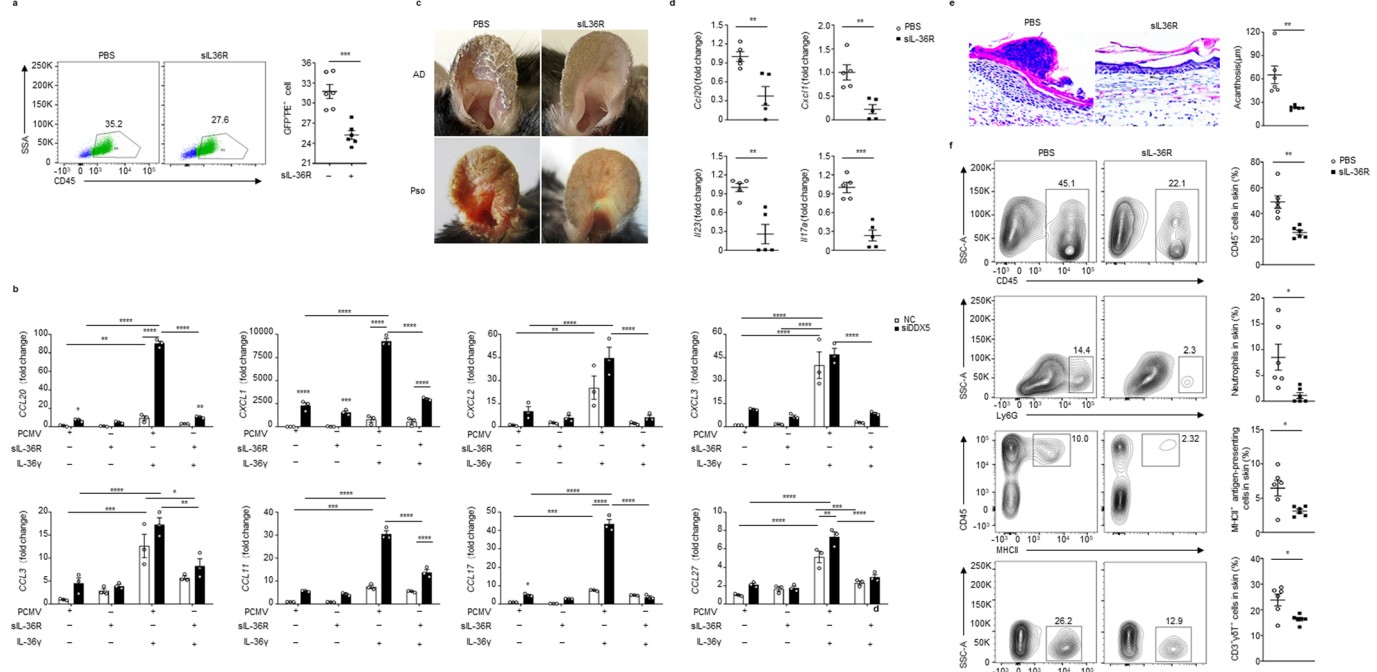

**Extended Data Fig. 9 | sIL-36R antagonizes IL-36R signaling to inhibit skin inflammation in keratinocyte, AD and psoriasis. a,** Flow cytometry analysis of the interaction between IL-36γ and IL-36R with or without sIL-36R overexpression in HeLa cells (*n* = 6). Data represent three independent experiments. **b,** RT-qPCR of *CCL20, CXCL1, CXCL2, CXCL3, CCL3, CCL11 CCL17* and *CCL27* in NHEKs in response to IL-36γ after DDX5 was silenced and/or sIL-36R was overexpressed. Data represent three independent experiments. **c,** Representative photos of ears from MC903-treated (*n* = 5) or IMQ-treated (*n* = 5) wild-type mice injected with PBS in left ears or 1 μg per ear recombinant sIL-36R in right ears. **d,** RT-qPCR of *Ccl20, Cxcl1, Il23* and *Il17a* in lesional skin from wild-type mice (*n* = 5) treated by IMQ as in (**b**). **e,** H&E staining of ear skin from wild-type mice treated by IMQ as in (**b**) and quantification of acanthosis of skin sections. **f,** Flow cytometry of CD45⁺ cells, CD11b⁺Ly6G⁺ neutrophils, CD11c⁺ APCs and CD3⁺γδTCR⁺ cells in ears from wild-type mice (*n* = 6) treated by IMQ as in (**b**). Data represent two independent experiments. *\*P* < 0.05, *\*\*P* < 0.01, *\*\*\*P* < 0.001 and *\*\*\*\*P* < 0.0001. *P* values were analyzed by two-way ANOVA (**b**) or unpaired, two-tailed Student's *t*-test (**a,d,e,f**).

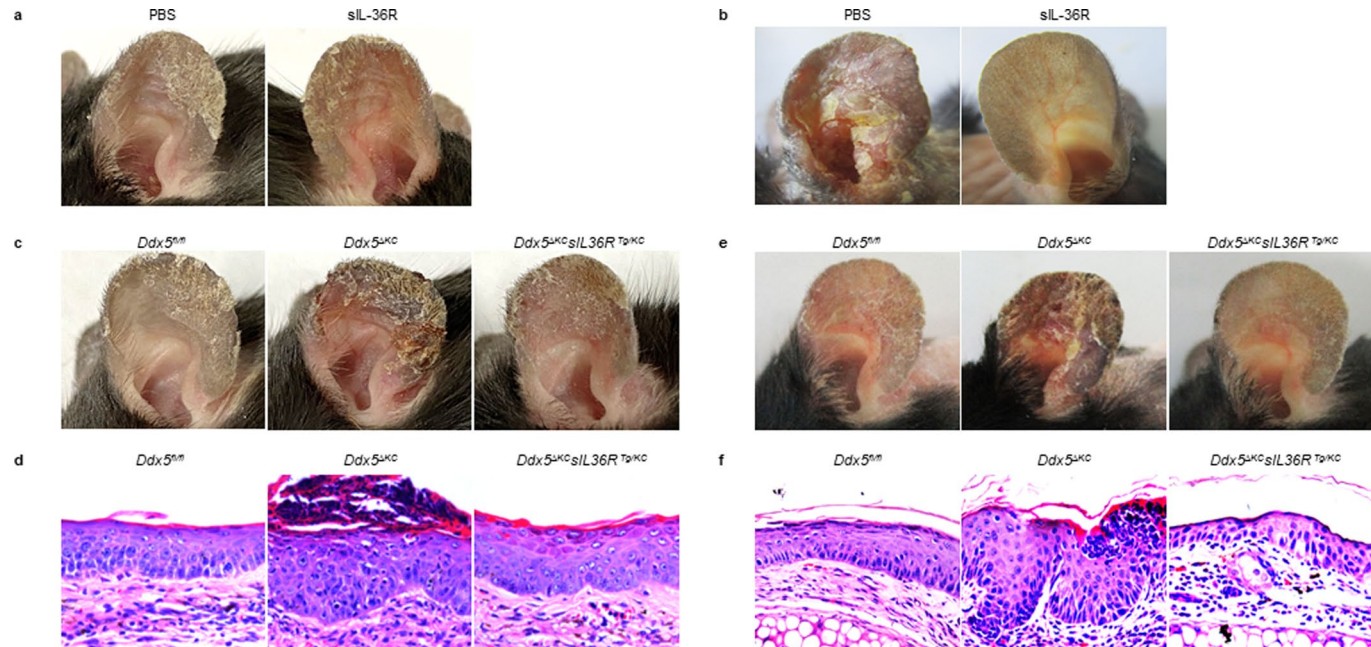

**Extended Data Fig. 10 | sIL-36R restoration inhibits inflammatory responses amplified by DDX5 deficiency. a-b**, Representative photos of ears from MC903-treated ($n = 4$) (**a**) or IMQ-treated ($n = 10$) $Ddx5^{\Delta KC}$ mice (**b**) with intradermal injection with PBS in left ears or 1 μg per ear recombinant sIL-36R in right ears. **c**, Representative photos of ears from MC903-treated $Ddx5^{fl/fl}$, $Ddx5^{\Delta KC}$ and

$Ddx5^{\Delta KC}sIL36R^{Tg/KC}$ mice ($n = 8$). **d**, H&E staining of ear skin from MC903-treated $Ddx5^{fl/fl}$, $Ddx5^{\Delta KC}$ and $Ddx5^{\Delta KC}sIL36R^{Tg/KC}$ mice ($n = 3$). **e**, Representative photos of ears from IMQ-treated $Ddx5^{fl/fl}$, $Ddx5^{\Delta KC}$ and $Ddx5^{\Delta KC}sIL36R^{Tg/KC}$ mice ($n = 8$). **f**, H&E staining of ear skin from IMQ-treated $Ddx5^{fl/fl}$, $Ddx5^{\Delta KC}$ and $Ddx5^{\Delta KC}sIL36R^{Tg/KC}$ mice ($n = 3$). Data represent at least two independent experiments.

# Reporting Summary

## Statistics

For all statistical analyses, confirm that the following items are present in the figure legend, table legend, main text, or Methods section.

| n/a | Confirmed | |
|---|---|---|
| ☐ | ☒ | The exact sample size (*n*) for each experimental group/condition, given as a discrete number and unit of measurement |
| ☐ | ☒ | A statement on whether measurements were taken from distinct samples or whether the same sample was measured repeatedly |
| ☐ | ☒ | The statistical test(s) used AND whether they are one- or two-sided<br>*Only common tests should be described solely by name; describe more complex techniques in the Methods section.* |
| ☐ | ☒ | A description of all covariates tested |
| ☐ | ☒ | A description of any assumptions or corrections, such as tests of normality and adjustment for multiple comparisons |
| ☐ | ☒ | A full description of the statistical parameters including central tendency (e.g. means) or other basic estimates (e.g. regression coefficient) AND variation (e.g. standard deviation) or associated estimates of uncertainty (e.g. confidence intervals) |
| ☐ | ☒ | For null hypothesis testing, the test statistic (e.g. *F*, *t*, *r*) with confidence intervals, effect sizes, degrees of freedom and *P* value noted<br>*Give P values as exact values whenever suitable.* |
| ☒ | ☐ | For Bayesian analysis, information on the choice of priors and Markov chain Monte Carlo settings |
| ☒ | ☐ | For hierarchical and complex designs, identification of the appropriate level for tests and full reporting of outcomes |
| ☒ | ☐ | Estimates of effect sizes (e.g. Cohen's *d*, Pearson's *r*), indicating how they were calculated |

*Our web collection on statistics for biologists contains articles on many of the points above.*

## Software and code

Policy information about availability of computer code

| Data collection | qRT-PCR was performed on a StepOnePlus™ Real-Time PCR System (Applied Biosystem).<br>RNA sequencing was performed on Hiseq (Illumina).<br>The processed RNA-Seq datasets for atopic dermatitis and psoriasis were downloaded from GEO database (GSE121212).<br>The processed scRNA-seq data was downloaded from doi: 10.5281/zenodo.4310074.<br>ELISA data were recorded with SPECTROstar Nano.<br>Flow cytometry data were collected on Fortessa (BD Biosciences).<br>The mass spectrometry resulting peptides were analyzed on the HPLC liquid system Dionex Ultimate 3000 (Thermo Scientific) coupled to a Dionex Trap column (100μm X 2cm X 5μm) with in-house packed C18 column (75μm X 15cm X 3μm) . |
|---|---|
| Data analysis | GraphPad Prism v9.0 was used for analyzing qRT-PCR, ELISA data and all the statistical analyses.<br>FastQC (version 0.11.9) was used for assessing the overall quality of raw reads and Trimmomatic (version 0.39) was applied for raw reads quality control to cut adapters and remove low-quality reads. And then all the clean reads were mapped onto human hg38 genome using Hisat2 (version 2.1.0). Samtools (version: 1.7) was used to convert SAM format files to BAM format files and sorted the BAM files. Gene expression levels were quantified by FeatureCounts (version 1.6.3). Differential expression analysis was performed by R package DESeq2 (Version 1.34.0). Gene ontology (GO) and KEGG pathway enrichment analyses were performed in ClusterProfiler (version 4.2.0). Differential alternative splicing events were detected by rMATS (version 3.1.0).<br>The processed scRNA-seq data was converted to Seurat object using SeuratDisk (https://mojaveazure.github.io/seurat-disk/).Followed by the typical Seurat workflow (http://satijalab.org/seurat/), 2000 highly variable genes were normalized, scaled and identified by using the NormalizeData, ScaleData and FindVariableGenes function from Seurat. The number of positive cells was determined by visual inspection of the ElbowPlot. Uniform Manifold Approximation and Projection (UMAP) with a resolution of 0.5 was used to determine cell clusters. The |

annotation information was reserved to define clusters. Cell subsets were combined and the expression level was visualized by VlnPlot from Seurat.

Flow cytometry data were analyzed by FLowJo v10.0 software (TreeStar) .

The mass spectrometry resulting peptides was analyzed by maXis HDTM-UHR-TOF mass spectrometer (Bruker). The spectra from mass spectrometry were automatically used for searching against the nonredundant International Protein Index human protein database (version 3.72) with the Bioworks browser (rev.3.1).

ESEfinder program (http://exon.cshl.edu/ESE) was used for evaluated the potential binding ESEs.

For manuscripts utilizing custom algorithms or software that are central to the research but not yet described in published literature, software must be made available to editors and reviewers. We strongly encourage code deposition in a community repository (e.g. GitHub). See the Nature Portfolio guidelines for submitting code & software for further information.

## Data

Policy information about availability of data

All manuscripts must include a data availability statement. This statement should provide the following information, where applicable:

- Accession codes, unique identifiers, or web links for publicly available datasets
- A description of any restrictions on data availability
- For clinical datasets or third party data, please ensure that the statement adheres to our policy

The raw sequence data reported in this paper have been deposited in Gene Expression Omnibus (GEO) Database under accession numbers GSE208666, GSE208669 and GSE208671. The mass spectrometry proteomics data have been deposited to the ProteomeXchange Consortium (http://www.proteomexchange.org) under the accession number  PXD021379. All other data supporting the findings of this study are available within the paper or from the corresponding author upon request.

# Field-specific reporting

Please select the one below that is the best fit for your research. If you are not sure, read the appropriate sections before making your selection.

☒ Life sciences ☐ Behavioural & social sciences ☐ Ecological, evolutionary & environmental sciences

For a reference copy of the document with all sections, see nature.com/documents/nr-reporting-summary-flat.pdf

# Life sciences study design

All studies must disclose on these points even when the disclosure is negative.

| | |
|---|---|
| Sample size | For all animal studies, we performed preliminary experiments to determine requirements for sample size, and at least twice independent experiments were performed to ensure reproducibility.  For all in vitro experiments, at least 3 independent biological replicates were used and three independent experiments were performed , with few exceptions in which experiments were repeated twice. |
| Data exclusions | In ELISA,the samples with low yield of protein were pre-determined and excluded. |
| Replication | All in vivo experiments were repeated at least twice, and in vitro experiments were conducted with three times with few exceptions in which experiments were repeated twice. Each replicated assay obtained the similar results. In each individual experiment, each technical replicate was measured once. |
| Randomization | Mice were grouped according to genotype and littermates were used for each animal experiment. |
| Blinding | All in vivo experiments were not performed in a blinded manner, because mice were grouped based on and compared across different genotypes, and different genotypes have distinct disease manifestation. However,we kept all experiments as unbiased as possible. Each psoriatic and AD model and measurement were performed by the same researcher to ensure reproducibility. Proper internal controls and normalization methods were included in each study forinternal bias. |

# Reporting for specific materials, systems and methods

We require information from authors about some types of materials, experimental systems and methods used in many studies. Here, indicate whether each material, system or method listed is relevant to your study. If you are not sure if a list item applies to your research, read the appropriate section before selecting a response.

## Materials & experimental systems

| n/a | Involved in the study |
|-----|------------------------|
| ☐ | ☒ Antibodies |
| ☐ | ☒ Eukaryotic cell lines |
| ☒ | ☐ Palaeontology and archaeology |
| ☐ | ☒ Animals and other organisms |
| ☐ | ☒ Human research participants |
| ☒ | ☐ Clinical data |
| ☒ | ☐ Dual use research of concern |

## Methods

| n/a | Involved in the study |
|-----|------------------------|
| ☒ | ☐ ChIP-seq |
| ☐ | ☒ Flow cytometry |
| ☒ | ☐ MRI-based neuroimaging |

# Antibodies

| Antibodies used | |
|---|---|
| | Antibodies for Western blot, Immunofluorescent staining and immunoprecipitation: |

Anti-DDDDK-tag: MBL, Cat #M185-3L, Clone # FLA-1, Lot#006,1:10000
Anti-HA-tag: MBL, Cat #M180-3S, Clone #TANA2, Lot#007,1:10000
Anti-His-tag: MBL, Cat #D291-3S, Clone #OGHIS, Lot#008,1:5000
Anti-β-Actin: Sigma, Cat #A5541, Clone #AC-15, Lot# 122M4782,1:5000
Anti-GAPDH: Proteintech, Cat #60004-1-Ig, Clone #Ag0766, Lot# 10013030 ,1:5000
Anti-DDX5: Abcam, Cat #ab126730, Clone #EPR7239, Lot# GR3273719-4,1:1000 for Immunoblotting,1:100 for Immunofluorescence
Anti-IL-17D: Thermo Fisher, Cat#MA5-24033, Clone# 312724, Lot#312724,1:500
Anti-SF2: Abcam, Cat #ab133689, Clone #EPR8240, Lot# GR97894-10,1:1000
Phospho-p38 MAPK (Thr180/ Tyr182) Antibody: CST, Cat #9211, Clone #D3F9, Lot#0022,1:1000
p38 MAPK Antibody: CST, Cat # 9212S, Clone #D3F9, Lot# 0023,1:1000
Phospho-AKT(Ser473): CST, Cat#4060, Clone #DE9, Lot#0012,1:1000
AKT(Pan) antibody: CST, Cat#4691, Clone #C67E7, Lot#0011,1:1000
Phospho-SMAD2(Ser465/467)/SMAD3(Ser423/425): CST, Cat#8828S, Clone #D27F4, Lot#008,1:1000
SMAD2/3 Antibody：CST, Cat#8685S, Clone #D7G7, Lot#007,1:1000
Phospho-NF-κB p65 (Ser536) (93H1) Rabbit Mab: CST, Cat#3033S, Clnoe#93H1, Lot#0017,1:1000
NF-κB p65 Rabbit mAB: CST, Cat #4764S, Clnoe#C22B4, Lot#0016,1:1000
Phospho-SAPK/JNK (Thr183/Tyr185)(81E11) Rabbit mAB: CST, Cat#4668S, Clone#81E11, Lot#0022,1:1000
Recombinant Anti-JNK2 Antibody, Rabbit monoclonal: Sino Biological, Cat#10745-R004, Clone#011, Lot#HB06SE1405,1:1000
Human IL-36 gamma/IL-1F9 Antibody: R&D, Cat#AF2320, Clone #Q9NZH8, Lot# UNN0112091,1:500
IL36γ Rabbit pAb: ABclonal, Cat # A10165, Clone #Q82460, Lot#Q2062,1:1000
anti-IL-36R antibody N-terminal mAb: Abcam,Cat#ab210933,Clone#ABM47A2, Lot#GR318274-1,1:1000
Mouse IL-1Rrp2/IL-1R6 Antibody: R&D, Cat#AF2354-SP, Clone# Q9ERS7, Lot# WVV0217081,1:1000
anti-Rabbit IgG: Abmart, Cat #B30011M, Lot#294670,1:20000
anti-Mouse IgG: Abmart, Cat #B30010M, Lot#294656,1:20000
Goat anti-Rabbit IgG (H+L) Cross-Adsorbed Secondary Antibody Alexa Fluor 488: Invitrogen, Cat #A-11008, Lot#1672238,1:1000
Goat anti-Mouse IgG (H+L) Cross-Adsorbed Secondary Antibody Alexa Fluor 488: Invitrogen, Cat#A-11011, Lot#2318440,1:1000
AffiniPure Donkey Anti-Mouse IgG (H+L): Jackson ImmunoResearch, Cat #715-005-151, Clone#AB_2340759, Lot#147635,1:10000
AffiniPure Goat Anti-Rabbit IgG (H+L): Jackson ImmunoResearch, Cat #111-005-003, Clone# AB_2337913, Lot#152677,1:10000
AffiniPure Donkey Anti-Goat IgG (H+L): Jackson ImmunoResearch, Cat#705-065-147, Clone#AB_2340385, Lot#119956,1:10000
Antibodies for FACS:
PE Anti-His: BioLegend, Cat #362603, Clone#J095G46, Lot#B226473,0.25μg/10^6 cell
Zombie Violet Dye: BioLegend, Cat #423113, Lot#B281932,1:1000
PE Anti-Mouse CD45: BioLegend, Cat #103105, Clone#30-F11, Lot#B294742,0.25μg/10^6 cell
APC Anti-Mouse CD45: BioLegend, Cat #103111, Clone#30-F11, Lot#B308253,0.25μg/10^6 cell
FITC Anti-Mouse CD45: BioLegend, Cat #103108, Clone#30-F11, Lot#B246762,0.25μg/10^6 cell
FITC Anti-Mouse CD3: BioLegend, Cat #100203, Clone#17A2, Lot#B313076,0.25μg/10^6 cell
APC Anti-Mouse γδTCR: BioLegend, Cat #118116, Clone#GL3, Lot#B228498,0.25μg/10^6 cell
PE Anti-Mouse CD11b: BioLegend, Cat #101207, Clone#M1/70, Lot#B253921,0.25μg/10^6 cell
FITC Anti-Mouse CD11b: BioLegend, Cat #101205, Clone#M1/70, Lot#B324795,0.25μg/10^6 cell
FITC Anti-Mouse Ly6G: BioLegend, Cat #108405, Clone#RB6-8C5, Lot#B281018,0.25μg/10^6 cell
APC Anti-Mouse CD11c: BioLegend, Cat #117309, Clone#N418, Lot#B297573,0.25μg/10^6 cell
PE Anti-Mouse CD11c: BioLegend, Cat #117308, Clone#N418, Lot#B234524,0.25μg/10^6 cell
PerCP/Cy5.5 Anti-Mouse IA-IE: BioLegend, Cat #107625, Clone#M5/114.15.2, Lot#B261237,0.25μg/10^6 cell
PE Anti-Mouse CD170: BioLegend, Cat #155505, Clone#S17007L, Lot#B301118,0.25μg/10^6 cell
PerCP/Cyanine5.5 Anti-Mouse CD49b: BioLegend, Cat #103519, Clone#HMα2, Lot#B294947,0.25μg/10^6 cell
FITC Anti-Mouse IgE: BioLegend, Cat #406905, Clone#RME-1, Lot#B316961,0.25μg/10^6 cell
PerCP/Cyanine5.5 Anti-Mouse CD4: BioLegend, Cat #100433, Clone#GK1.5, Lot#B248433,0.25μg/10^6 cell
Antibodies for ELISA:
Mouse Il13 ELISA Ready-SET-Go: eBioscience, Cat#88-7137-88, Lot#E09414
Mouse Tslp ELISA Kit: MULTI SCIENCES, Cat#EK265/2-96, Lot#A26590752
Mouse Il4 ELISA Kit: MULTI SCIENCES, Cat#EK204/2-96, Lot#A20400253
DuoSet mouse Ccl20: R&D, Cat#DY760, Lot#P167295
DuoSet mouse Cxcl1: R&D, Cat#DY453-05, Lot#P179973
DuoSet mouse Il23: R&D, Cat#DY1887-05, Lot#P190436
DuoSet mouse Il17a: R&D, Cat#DY421-05, Lot#P328895
Mouse Tnfα: BD Pharmingen, Cat#51-9004717, Lot#10211
Mouse Il17f: BD Pharmingen, Cat#562174, Lot#8162567

| Validation | Other antibodies:<br>Anti-Mouse sIL36R &Anti-Human sIL36R made by Sirtomics Biotechnology Company who was entrusted by our laboratory. |
| | All above commercial antibodies are well validated by the manufacturer. Specificity and validation were provided by manufacturer's technical datasheets and confirmed in literature.Please refer to the spec sheets on the respective vendors' websites for technical information and detail by searching with the catalog numbers provided. For Anti-Mouse sIL36R &Anti-Human sIL36R antibody, we did western blotting to test their specificity. |

# Eukaryotic cell lines

Policy information about cell lines

| Cell line source(s) | HEK293T cells and HeLa cells were provided by Prof. Wong from East China Normal University who purchased these cell lines from ATCC, HaCaT cells were purchased from Cobioer. |
| Authentication | HEK293T cell line and HeLa cells were not tested and authenticated by our laboratory. HaCaT cell line has been tested and authenticated, using morphology, karyotyping and PCR based approaches to confirm the identity by the vendor. |
| Mycoplasma contamination | All cell lines were tested for mycoplasma contamination and were confirmed negative. |
| Commonly misidentified lines<br>(See ICLAC register) | No commonly misidentified cell lines were used. |

# Animals and other organisms

Policy information about studies involving animals; ARRIVE guidelines recommended for reporting animal research

| Laboratory animals | Mice with BABL/c or C57BL/6 background were bred in the specific-pathogen-free animal facility at East China Normal University.Il17d–/– and CD93–/– mice were bred in the specific-pathogen-free animal facility at Tsinghua University. DDX5fl/fl and sIL36R fl/fl mice were generated by Shanghai Model Organisms Center, Inc. The K14Cre transgenic mice were obtained from Shanghai Model Organisms Center, Inc, while the K5Cre transgenic mice were obtained from Xiao Yang lab in the Academy of Military Medical Sciences in China. DDX5fl/fl mice were crossed with the K14Cre transgenic mice to generate Ddx5ΔKC mice, sIL36Rfl/fl mice were crossed with the K5Cre mice to generate sIL36RTg/KC mice, and Ddx5ΔKC mice were crossed with sIL36RTg/KC mice to generate Ddx5ΔKCsIL36RTg/KC mice. Both male and female mice aged at 7-8 weeks old were used in all experiments. |
| Wild animals | No wild animals were used in this study. |
| Field-collected samples | No field-collected samples were used in this study. |
| Ethics oversight | All the animal experiments were performed with the use of the protocols (Protocol No.: m20210233 for AD, m20200316 for psoriasis) approved by the Animal Care and Use Committee at East China Normal University. |

Note that full information on the approval of the study protocol must also be provided in the manuscript.

# Human research participants

Policy information about studies involving human research participants

| Population characteristics | Patients with atopic dermatitis: moderate to severe, 3 men and 4 women; patients with psoriasis: mild plaque-type psoriasis, 4 men and 6 women; patients with basal cell carcinoma:1 man and 2women; patients with squamous cell carcinoma:2 men and 1woman; normal patients: 4 men and 4 women. Patients' age are from 14-80 years old. |
| Recruitment | Before enrolling in the study, all patients were volunteers and aware of the subjects of the study and signed the informed consent. All patients are from Shanghai, China. The only criteria for inclusion were healthy, AD, psoriasis, SCC or BCC. |
| Ethics oversight | The Ethics Review Committees of Huashan Hospital or Shanghai Tenth People's Hospital approved the protocols (KY2020732 for AD, SHSY-IEC-KY-4.0/18-13/01 for psoriasis)used in this study. |

Note that full information on the approval of the study protocol must also be provided in the manuscript.

# Flow Cytometry

## Plots

Confirm that:

☒ The axis labels state the marker and fluorochrome used (e.g. CD4-FITC).

☒ The axis scales are clearly visible. Include numbers along axes only for bottom left plot of group (a 'group' is an analysis of identical markers).

☒ All plots are contour plots with outliers or pseudocolor plots.

☒ A numerical value for number of cells or percentage (with statistics) is provided.

# Methodology

| | |
|---|---|
| Sample preparation | For the skin samples, the epidermis and dermis were separated using dispase II (Sigma; 5mg/mL in HBSS; 37°C for 90 min). The dermal cells were separated by collagenase (Roche) and hyaluronidase (Sigma) digestion [10mM Hepes, collagenase D (2.5mg/mL), hyaluronidase (100U/mL), and deoxyribonuclease (50μg/mL) in DMEM (GIBICO); 37°C for 45 min]. Isolated cells were stained with different cell surface markers [CD45 for leukocytes; CD45+, CD49b+,IgE+ for basophils; CD45+, CD11b+, SiglecF+ for eosinophils; CD45+, CD3+, CD4+ for CD4+ T cells; CD45+, CD11b+, Ly6G+ for neutrophils; CD45+, CD11c+ or MHCII+ for antigen presenting cells; CD45+, CD3+, γδT cell receptor (TCR+) for γδT cells]. The cells were then fixed and the relevant isotype control mAbs were used. |
| Instrument | BD LSR Fortessa |
| Software | FlowJo v10 |
| Cell population abundance | All frequencies of cells are stated in the representative graphs and quantifications. |
| Gating strategy | Cells were first gated based on FSC-A and SSC-A to exclude debris, and then gated by live-dead dye to exclude dead cells. Specific cell populations were determined by markers listed as follows: CD45 for leukocytes; CD45+ CD11b+Ly6G+ for neutrophils; CD45+CD11c+ or MHCII+ for dendritic cells; CD45+CD3+γδT cell receptor (TCR+) for γδT cells; CD45+ ,CD49b+, IgE+ for basophils; CD45+CD11b+ SiglecF+ for eosinophils; CD45+ CD3+CD4+ for CD4+ T cells. |

☒ Tick this box to confirm that a figure exemplifying the gating strategy is provided in the Supplementary Information.

nature portfolio | reporting summary

March 2021

30