## [Peer Review File · Nature Immunology]

Peer Review Information

Journal: Nature Immunology

Manuscript Title: IL-17D-induced inhibition of DDX5 expression in keratinocytes amplifies IL-36R-mediated skin inflammation

Corresponding author name(s): Yuping Lai

Reviewer Comments & Decisions:

Decision Letter, initial version:
--

Subject: Decision on Nature Immunology submission NI-A32987

Message: 10th Dec 2021

Dear Dr. Lai,

Your Article, "IL-17D-induced keratinocyte DDX5/sIL-36R disorder drives skin inflammation" has now been seen by 2 referees. While we find your work of considerable potential interest, the reviewers have raised substantial concerns that must be addressed. As such, we cannot accept the current version of the manuscript for publication, but would be happy to consider a revised version that addresses these concerns, as long as novelty is not compromised in the interim.

Please revise the manuscript to address all issues raised by the referees. We consider it is important to better focus the manuscript around the most high-impact aspect of the work, which is that altered splicing is driving the phenotype observed. The referees thought the manuscript was too broad and it would benefit from better focusing to clearly present and discuss the data most important for the main conclusions of the paper.

At resubmission, please include a point-by-point "Response to referees" detailing how you have addressed each referee comment (please specify page and figure number where the new data can be found in the revised manuscript). This response will be sent back to the referees along with the revised manuscript.

In addition, please include a revised version of any required reporting checklist. It will be available to referees (and, potentially, statisticians) to aid in their evaluation if the manuscript goes back for peer review. A revised checklist is essential for re-review of the paper.

When submitting the revised version of your manuscript, please pay close attention to our

[href="https://www.nature.com/nature-research/editorial-policies/image-integrity">Digital Image Integrity Guidelines.](https://www.nature.com/nature-research/editorial-policies/image-integrity) and to the following points below:

You may use the link below to submit your revised manuscript and related files:
[REDACTED]

We hope to receive the revised manuscript within 6 months. If you cannot send it within this time, please let us know. We will be happy to consider your revision so long as nothing similar has been accepted for publication at Nature Immunology or published elsewhere.

Nature Immunology is committed to improving transparency in authorship. As part of our efforts in this direction, we are now requesting that all authors identified as 'corresponding author' on published papers create and link their Open Researcher and Contributor Identifier (ORCID) with their account on the Manuscript Tracking System (MTS), prior to acceptance. ORCID helps the scientific community achieve unambiguous attribution of all scholarly contributions. You can create and link your ORCID from the home page of the MTS by clicking on 'Modify my Springer Nature account'. For more information please visit www.springernature.com/orcid.

Thank you for the opportunity to review your work.

Sincerely,

Ioana Visan, Ph.D.
Senior Editor
Nature Immunology

Tel: 212-726-9207
Fax: 212-696-9752

www.nature.com/ni

Reviewers' Comments:

Reviewer #1:

Remarks to the Author:

This manuscript by Ni et. al investigates the role of the RNA helicase DDX5 in atopic dermatitis and psoriasis. The group discovers that DDX5 is significantly downregulated in skin affected by AD or psoriasis, and that DDX5 knockout mice develop skin disease in MC903 and IMQ induction models. Subsequently, they determine that excess IL-17D, an inflammatory cytokine associated with both models of skin disease, causes downregulation DDX5. The authors next investigate the mechanism by which DDX5 deficiency drives skin pathology and discover that its absence leads to alternative splicing of the IL36 receptor. When DDX5 is downregulated, keratinocytes produce significantly more of the membrane-bound isoform of the receptor, IL36R, and little to no sIL36R, a soluble form that acts as a decoy receptor. DDX5's direct interaction with SF2, a splice factor, is shown to enhance SF2 binding with exon 4 of IL36R, causing splice machinery to skip exon 3 and thus generate sIL36R. Ni et. al then demonstrate that intradermally administered sIL36R is sufficient to prevent inflammation, as well as restore DDX5-deficient mice to WT levels of skin pathology in both AD and psoriasis models.

This manuscript convincingly proves that DDX5 complexes with the splice factor SF2 to modulate splicing of the IL36 receptor, thereby ensuring that inflammatory cytokines do not signal in excess, causing skin pathology. The one thing that remains to be seen is the connection between IL-17D and DDX5 reduction, which was not clear. However, this is a significant finding with relevance to human health.

Major point:

1) Much of the investigation in this paper relies on the idea that DDX5 deficiency causes changes in global splicing. The authors state, "quantification of splicing events from the RNA-Seq data demonstrated that DDX5 deficiency changed the splicing patterns in keratinocyte. Therefore, we hypothesized that DDX5 might regulate IL-36R pre-mRNA splicing to control IL-36R expression" (lines 227-230). However, the authors do not show any data for the splicing analysis or how quantification of splicing events from the RNA-Seq was conducted. We would like to see the interpretation of the RNA seq data in DDX5 deficient mice that shows aberrant splicing compared to WT controls and how the sequencing data was analyzed.

2) Was only IL-36R significantly altered? Where there other genes in the inflammatory pathway that could also be affected by DDX?

3) In Figure 3d, knockdown of the CD93 receptor caused abrogation of the DDX5 downregulation phenotype seen with IL-17D treatment. However, in the discussion section, the authors claim that IL-17D did not activate CD93 signal transduction or change IL-36R splicing and do not show the data. These data need to be included in the supplement and a proposal for alternative CD93 signaling that could cause this phenomenon.

Minor points:

- 1) The methodology and experimental setup of figure 5j in which the authors determine Exon 3 versus Exon 4 binding to IL-36R is confusing and unclear based on the current written explanation. Further clarification of the assay in the text is required.
- 2) Gene ontology analysis for Fig 5A should be presented.
- 3) Validation of DDX siRNA knockdowns (Fig 5B) have not been provided. Protein or mRNA quantification of knock-down efficiency is critical to assess these data.
- 4) Accession number (for RNA-seq) provided links to unrelated datasets.

Reviewer #2:

Remarks to the Author:

In this manuscript by Ni and colleagues, the authors report a highly complex mechanism involving IL-17D, CD93, DDX5 and sIL36R that underlies cutaneous inflammation. Based on genomic analyses of psoriasis and atopic dermatitis, they focus on the DDX5's role in cutaneous inflammation. They nicely demonstrate that DDX5 expression is reduced in the setting of cutaneous inflammation and that a KC-specific ablation of DDX5 attenuates inflammation in multiple models of cutaneous inflammation. They go on to show that IL-17D from an unidentified source inhibits inflammation-induced KC expression of DDX5 through a CD93-dependent mechanism. By screening a variety of likely suspects, they found that DDX5 suppresses the response of KC to IL36g. Using a transcriptomic approach, they focus onto KC expression of IL36R as the key DDX5-dependent process. Impressively, they demonstrate that DDX5 is critical for exon skipping of IL-36R processing to produce the soluble and inhibitory sIL36R through a mechanism that involves SF2. Finally, they demonstrate that sIL36R antagonized IL36R signaling thereby explaining the inflammation seen in DDX5 deficiency. The breadth of this manuscript is breath taking and the findings quite impressive and overall convincing. My chief concern is that the breadth of the manuscript required fairly light discussion of each of the individual sections. This makes proper understanding of the methods employed challenging to decipher. There also are some important aspects that are not explored. For instance, in figure 3E, the authors note the large differences in gene expression comparing WT vs DDX5^{-/-} cells following exposure to cytokines. The authors use this to justify focusing on IL36g, however, there are large differences in TNF α , IL25, and IL4 which may only appear inconsequential due to the very high expression of IL36g in WT mice. In addition, the relationship between IL17D and CD93 in this model is somewhat overlooked. It may not be key, but it is a glaring hole in an otherwise extensive model. Key concerns:

- 1- The manuscript is simply too broad. The authors may wish to remove some data to allow for deeper discussion of each component of the model. Perhaps the section on SF2 is somewhat out of place.
- 2- If DDX5 control splicing of IL36R then why are KC responses to other cytokines also affected? Is there a similar or different mechanism. Examining this issue would greatly broaden the impact of the finding.
- 3- How does receptor for IL17D fit into this story?
- 4- Please include a model figure. This is really important.

Minor comments.

- 5- Figure 1 uses MC903 but figure 2 focuses on IMQ. Are results the same?
 6- Figure 1j, are fibroblasts also affected?
 7- Labels in Figure 3e-f unclear.
 8- Figure 1g. Please include stats. Is expression reduced in uninvolved skin? Why would this be the case?

Author Rebuttal to Initial comments

Point-by-point response

Reviewer #1:

Remarks to the Author:

This manuscript by Ni et. al investigates the role of the RNA helicase DDX5 in atopic dermatitis and psoriasis. The group discovers that DDX5 is significantly downregulated in skin affected by AD or psoriasis, and that DDX5 knockout mice develop skin disease in MC903 and IMQ induction models. Subsequently, they determine that excess IL-17D, an inflammatory cytokine associated with both models of skin disease, causes downregulation DDX5. The authors next investigate the mechanism by which DDX5 deficiency drives skin pathology and discover that its absence leads to alternative splicing of the IL36 receptor. When DDX5 is downregulated, keratinocytes produce significantly more of the membrane-bound isoform of the receptor, IL36R, and little to no sIL36R, a soluble form that acts as a decoy receptor. DDX5's direct interaction with SF2, a splice factor, is shown to enhance SF2 binding with exon 4 of IL36R, causing splice machinery to skip exon 3 and thus generate sIL36R. Ni et. Al then demonstrate that intradermally administered sIL36R is sufficient to prevent inflammation, as well as restore DDX5-deficient mice to WT levels of skin pathology in both AD and psoriasis models.

This manuscript convincingly proves that DDX5 complexes with the splice factor SF2 to modulate splicing of the IL36 receptor, thereby ensuring that inflammatory cytokines do not signal in excess, causing skin pathology. The one thing that remains to be seen is the connection between IL-17D and DDX5 reduction, which was not clear. However, this is a significant finding with relevance to human health.

Major point:

Comment 1) *Much of the investigation in this paper relies on the idea that DDX5 deficiency causes changes in global splicing. The authors state, "quantification of splicing events from the RNA-Seq data demonstrated that DDX5 deficiency changed the splicing patterns in keratinocyte.*

Therefore, we hypothesized that DDX5 might regulate IL-36R pre-mRNA splicing to control IL-36R expression” (lines 227-230). However, the authors do not show any data for the splicing analysis or how quantification of splicing events from the RNA-Seq was conducted. We would like to see the interpretation of the RNA seq data in DDX5 deficient mice that shows aberrant splicing compared to WT controls and how the sequencing data was analyzed.

Response: As the reviewer suggested, we conducted RNA sequencing on the lesional skin from *Ddx5^{Δ/kc}* and *Ddx5^{fl/fl}* AD- and psoriasis-like mice and analyzed differential gene expression by R package DESeq2 (Version 1.34.0). We found that the expression of chemokines and cytokines related to the disease pathology was also increased in the lesional skin of *Ddx5^{Δ/kc}* AD- and psoriasis-like mice compared to their controls (Extended Data Fig. 5f,g, page 10 in the revised version). Moreover, we did Gene ontology (GO) enrichment analysis of RNA-Seq datasets from WT and *DDX5^{-/-}* HaCaT, *Ddx5^{Δ/kc}* and *Ddx5^{fl/fl}* AD- or psoriasis-like mice in ClusterProfiler (version 4.2.0) and found that DDX5-mediated changes in differential gene expression mostly affected leukocyte migration, positive regulation of cytokine production and cell chemotaxis in keratinocytes, AD and psoriasis (Fig.4b, page 10 in the revised version), which is consistent with the observation that cutaneous inflammation in *Ddx5^{Δ/kc}* AD- and psoriasis-like mice was much more severe than that in *Ddx5^{fl/fl}* AD- and psoriasis-like mice. Furthermore, we evaluated differential alternative splicing events changed in WT and *DDX5^{-/-}* HaCaT and in the lesional skin of *Ddx5^{Δ/kc}* and *Ddx5^{fl/fl}* AD- or psoriasis-like mice by rMATS (version 3.1.0), and observed that multiple splicing events, including skipped exon(SE), mutually exclusive exon (MXE), alternative 5' splice site (A5SS), alternative 3' splice site (A3SS) and retained intron (RI), were changed in *DDX5^{-/-}* HaCaT and the lesional skin of *Ddx5^{Δ/kc}* AD- or psoriasis-like mice compared to their controls. In particular, the event of skipped exon was most profoundly changed in *DDX5^{-/-}* HaCaT and the lesional skin of *Ddx5^{Δ/kc}* AD- or psoriasis-like mice (Extended Data Fig. 6a,b, page 12 in the revised version).

Comment 2) *Was only IL-36R significantly altered? Where there other genes in the inflammatory pathway that could also be affected by DDX?*

Response: No, DDX5 affected the alternative splicing pattern of several genes. For example, the splicing patterns of IL-7R, CD6, FGFR2 and IL-20RB were also markedly changed in *DDX5^{-/-}* HaCaT (Extended Data Fig. 6b, page 12 in the revised version).

Comment 3) *In Figure 3d, knockdown of the CD93 receptor caused abrogation of the DDX5 downregulation phenotype seen with IL-17D treatment. However, in the discussion section, the authors claim that IL-17D did not activate CD93 signal transduction or change IL-36R splicing*

and do not show the data. These data need to be included in the supplement and a proposal for alternative CD93 signaling that could cause this phenomenon.

Response: Thank you very much for the reviewer's critique. This critique prompted us to further explore IL-17D/CD93-mediated signaling pathway. To determine the signaling downstream of IL-17D/CD93, we expanded our screening library and used multiple inhibitors of potential CD93-mediated pathways to treat neonatal human epidermal keratinocytes (NHEKs) in the presence or absence of IL-17D, and found that among these inhibitors, p38 MAPK inhibitor, AKT1/2 inhibitor and SMAD2/3 inhibitor abrogated the inhibitory effect of IL-17D on both mRNA and protein expression of DDX5 (Fig. 3d, e). We also confirmed that IL-17D can induce phosphorylation of p38MAPK, AKT and SMAD2/3 in NHEKs or in the lesional skin of WT AD- and psoriasis-like mice, but this effect was lost in NHEKs with CD93 silencing or in the lesional skin of *Cd93*^{-/-} AD- and psoriasis-like mice (Fig. 3f, g and Extended Data Fig. 4j). Moreover, we have observed that p38MAPK inhibitor dampened IL-17D-induced AKT and SMAD2/3 phosphorylation while AKT1/2 and SMAD2/3 inhibitors did not have the capacity to block IL-17D-induced p38 MAPK phosphorylation in NHEKs (Fig. 4h-j). Furthermore, AKT1/2 inhibitor prevented IL-17D-induced SMAD2/3 phosphorylation, but not vice versa (Fig. 4i, j). All these data demonstrate that IL-17D activates CD93-p38MAPK-AKT-SMAD2/3 signaling pathway to inhibit DDX5 in keratinocytes and in the skin of AD and psoriasis. We have added all these data in Page 8-9 in the revised version.

Minor points:

Comment 1) *The methodology and experimental setup of figure 5j in which the authors determine Exon 3 versus Exon 4 binding to IL-36R is confusing and unclear based on the current written explanation. Further clarification of the assay in the text is required.*

Response: Figure 5j is Figure 5i in this revised version. As the reviewer suggested, we added more details of the assay in Figure legend as follows: IL-36R-reporter minigene, exogenous Flag-tagged SF2 and/or HA-tagged DDX5 were expressed in *DDX5*^{-/-} or *SF2*^{-/-} HeLa cells, respectively. Nuclear extracts prepared from these cells were used for immunoprecipitation with anti-Flag beads or anti-HA beads. Total RNA from RNP (Messenger ribonucleoprotein) components binding on beads was isolated and reversely transcribed into cDNA. Exon 3 or exon 4 binding to SF2 or DDX5 was amplified by PCR and detected by agarose gel. Lanes 1&2: SF2-binding exon 3 or exon 4 in nuclear extracts immunoprecipitated with anti-Flag beads. Lanes 3&4 (loading controls of Lane 1&2): exon 3 or exon 4 in total nuclear extracts. Lane 5&6: SF2-binding exon 3 or exon 4 in nuclear extracts immunoprecipitated with anti-HA beads. Lanes 7&8 (loading controls of Lane5&6): exon 3 or exon 4 in total nuclear extracts (Page 42-43 in the revised version).

Comment 2) *Gene ontology analysis for Fig 4A should be presented.*

Response: As suggested by the reviewer, we have presented gene ontology analysis of the original Fig. 4a in Fig. 4b and also did GO analysis of RNA-Seq datasets for *Ddx5*^{Δ/kc} and *Ddx5*^{fl/fl} AD- and psoriasis-like mice (Fig.4b, Page 10 in the revised version)

Comment 3) *Validation of DDX siRNA knockdowns (Fig 5B) have not been provided. Protein or mRNA quantification of knock-down efficiency is critical to assess these data.*

Response: As the reviewer suggested, we showed the data of validation of DDX knock-down efficiency in Extended data Fig.6g (Page 12 in the revised version)

Comment 4) *Accession number (for RNA-seq) provided links to unrelated datasets.*

Response: We apologized to show Submission ID as Accession number in the first version. In this revised version, we provide accession numbers in the section of “Data and Code Availability” in page 33. The accession numbers are HRA000306, CRA006953 and CRA006952 that are publicly accessible at <http://bigd.big.ac.cn/gsa-human>.

Reviewer #2:

Remarks to the Author:

In this manuscript by Ni and colleagues, the authors report a highly complex mechanism involving IL-17D, CD93, DDX5 and sIL36R that underlies cutaneous inflammation. Based on genomic analyses of psoriasis and atopic dermatitis, they focus on the DDX5's role in cutaneous inflammation. They nicely demonstrate that DDX5 expression is reduced in the setting of cutaneous inflammation and that a KC-specific ablation of DDX5 attenuates inflammation in

multiple models of cutaneous inflammation. They go on to show that IL-17D from an unidentified source inhibits inflammation-induced KC expression of DDX5 through a CD93-dependent mechanism. By screening a variety of likely suspects, they found that DDX5 suppresses the response of KC to IL36g. Using a transcriptomic approach, they focus onto KC expression of IL36R as the key DDX5-dependent process. Impressively, they demonstrate that DDX5 is critical for exon skipping of IL-36R processing to produce the soluble and inhibitory sIL36R through a mechanism that involves SF2. Finally, they demonstrate that sIL36R antagonized IL36R signaling thereby explaining the inflammation seen in DDX5 deficiency. The breadth of this manuscript is breath taking and the findings quite impressive and overall convincing. My chief concern is that the breadth of the manuscript required fairly light discussion of each of the individual sections. This makes proper understanding of the methods employed challenging to decipher. There also are some important aspects that are not explored. For instance, in figure 3E, the authors note the large differences in gene expression comparing WT vs DDX5^{-/-} cells following exposure to cytokines. expression. The authors use this to justify focusing on IL36g, however, there are large differences in TNF α , IL25, and IL4 which may only appear inconsequential due to the very high expression of IL36g in WT mice. In addition, the relationship between IL17D and CD93 in this model is somewhat overlooked. It may not be key, but it is a glaring hole in an otherwise extensive model.

Key concerns:

Comment 1- *The manuscript is simply too broad. The authors may wish to remove some data to allow for deeper discussion of each component of the model. Perhaps the section on SF2 is somewhat out of place.*

Response: Thank you for the reviewer's nice suggestion! Given that the SF2 data provides information on the molecular basis of how DDX5 regulates IL-36R splicing, we kept this finding in the manuscript, but moved some data to Extended data Fig.7 and presented the data more concisely (in page 13-15 in the revised version) after we discussed with the editor.

Comment 2- *If DDX5 control splicing of IL36R then why are KC responses to other cytokines also affected? Is there a similar or different mechanism. Examining this issue would greatly broaden the impact of the finding.*

Response: To explore the mechanism involved in DDX5^{-/-} keratinocytes response to IL-17A, TNF α , IL-25 and IL-4, we first evaluated the effect of TNF α , IL-4 and IL-25 on the expression of DDX5, IL-36R and IL-36 γ , and found that TNF α , but not IL-4 and IL-25, increased the

expression of IL-36R and IL-36 γ in keratinocytes (See Figure 1 attached to this point-by-point-response). In addition to TNF α , IL-17A has been reported to induce IL-36 γ in keratinocytes¹. All these findings suggest that the response of *DDX5*^{-/-} keratinocytes to IL-17A and TNF α is probably due to IL-17A- or TNF α -induced IL-36R and IL-36 γ . Since IL-4 and IL-25 did not affect the expression of *DDX5*, IL-36R and IL-36 γ , we next tested whether *DDX5* would regulate the splicing change of IL-4R (a receptor for IL-4) and IL-17RA (a receptor for IL-17A and IL-25). *DDX5* deficiency had slight effect on the splicing of IL-4R and IL-17RA in keratinocytes (Extended data Fig.6b), which was consistent with slight increased responses of *DDX5*^{-/-} keratinocytes to IL-4 and IL-25. Of course, there might be other mechanisms involved in *DDX5*^{-/-} keratinocyte response to IL-4 and IL-25, which needs further investigation, but out of the scope of the current manuscript.

Comment 3- *How does receptor for IL17D fit into this story?*

Response: Thank you for the great question! To address this, we first confirmed the role of CD93 in IL-17D inhibiting *DDX5* by developing WT and *Cd93*^{-/-} AD- and psoriasis-like mice. We found that *DDX5* was markedly increased in the lesional skin of *Cd93*^{-/-} AD- and psoriasis-like mice compared to that in WT AD- and psoriasis-like mice (Fig.3a and Extended Data Fig. 4f), which was accompanied by alleviated disease phenotypes and less expression of cytokines and chemokines (Fig.3b and Extended Data Fig. 4g,h). We then further explored IL-17D/CD93-mediated signaling pathway. To determine the signaling downstream of IL-17D/CD93, we expanded our screening library and used multiple inhibitors of CD93-mediated pathways to treat neonatal human epidermal keratinocytes (NHEKs) in the presence or absence of IL-17D, and found that among these inhibitors, p38 MAPK inhibitor, AKT1/2 inhibitor and SMAD2/3 inhibitor abrogated the inhibitory effect of IL-17D on both mRNA and protein expression of *DDX5* (Fig. 3d, e). We also confirmed that IL-17D can induce phosphorylation of p38MAPK, AKT and SMAD2/3 in NHEKs or in the lesional skin of WT AD- and psoriasis-like mice, but this effect was lost in NHEKs with CD93 silencing or in the lesional skin of *Cd93*^{-/-} AD- and psoriasis-like mice (Fig. 3f, g and Extended Data Fig. 4j). Moreover, we have observed that p38MAPK inhibitor dampened IL-17D-induced AKT and SMAD2/3 phosphorylation while AKT1/2 and SMAD2/3 inhibitors did not have the capacity to block IL-17D-induced p38 MAPK phosphorylation in NHEKs (Fig. 4h-j). Furthermore, AKT1/2 inhibitor prevented IL-17D-induced SMAD2/3 phosphorylation, but not vice versa (Fig. 4i, j). All these data demonstrate that IL-17D activates CD93-p38MAPK-AKT-SMAD2/3 signaling pathway to inhibit *DDX5* in keratinocytes and in the skin of AD and psoriasis. All the data have been added in Page 8-9 in this revised version.

Comment 4- Please include a model figure. This is really important.

Response: As the reviewer suggested, we added a model figure in Extended data Fig.10 to summarize our finding (page 18 in the revised version).

Minor comments.

Comment 5- Figure 1 uses MC903 but figure 2 focuses on IMQ. Are results the same?

Response: In both Figure 1 and 2, we have evaluated the expression and role of DDX5 in both MC903-induced AD-like mice and IMQ-induced psoriasis-like mice. In this revised version, we have moved the data that present the role of IMQ-induced psoriasis-like mice to Extended data Fig.2.

Comment 6- Figure 1j, are fibroblasts also affected?

Response: DDX5 seems to be increased in the dermis at day 4 in MC903-induced AD-like mice or at Day 1 in IMQ-induced psoriasis-like mice (Fig. 1j,k). However, when we used MC903 and IMQ to stimulate murine fibroblasts, we did not observe that MC903 or IMQ markedly affected both mRNA and protein levels of DDX5 in fibroblasts (See Figure 2 attached to this point-by-point response). Moreover, IL-17D that significantly inhibits DDX5 expression in keratinocytes cannot inhibit DDX5 in fibroblasts (See Figure 2 attached to this point-by-point response).

Comment 7- Labels in Figure 3e-f unclear.

Response: Sorry for that! We have restructured figures and Fig. 3e-f are now Fig. 2d-f in this revised version.

Comment 8- Figure 1g. Please include stats. Is expression reduced in uninvolved skin? Why would this be the case?

Response: As the reviewer suggested, we added stats in Fig. 1g. Yes, the expression of DDX5 mRNA was reduced in uninvolved skin. This is probably because unlesional skin was adjacent to lesional skin and IL-17D in lesional skin can also inhibit DDX5 expression in unlesional skin. However, although DDX5 mRNA was significantly reduced in unlesional skin in Fig.1g, we did

not observe that DDX5 was markedly decreased in unlesional skin from patients with psoriasis in Fig. 1i, which might be due to different distances of unlesional skin to lesional skin between our samples and samples in Fig. 1g or samples from patients with different races.

Figure 1. The expression of DDX5, IL-36R, IL-36 γ in keratinocytes response to IL-17E (IL-25), IL-4 and TNF α .

Figure 2. DDX5 expression in murine fibroblasts after stimulation with MC903, IMQ or IL-17D.

Reference

- Ovesen, S.K., Schulze-Osthoff, K., Iversen, L. & Johansen, C. IκBzeta is a Key Regulator of Tumour Necrosis Factor-α and Interleukin-17A-mediated Induction of Interleukin-36γ in Human Keratinocytes. *Acta Derm Venereol* **101**, adv00386 (2021).

Decision Letter, first revision:

Subject: Decision on Nature Immunology submission NI-A32987A

Message: Dear Dr. Lai,

The reviewers' comments on your manuscript "IL-17D-induced keratinocyte DDX5/sIL-36R disorder drives skin inflammation" are in. We are happy to inform you that if you revise your manuscript appropriately in response to the referees' comments and our editorial requirements your manuscript should be publishable in Nature Immunology.

Please revise your manuscript according with the reviewers' comments. At resubmission, please include a point-by-point response to the referees' comments, noting the pages and lines where the changes can be found in the revision. Please highlight the changes in the revised manuscript as well. Please note that articles for Nature Immunology have a word limit of 4000 words for the introduction, results and discussion and a limit of 50 references for the main text. The discussion should not exceed 800 words.

We are trying to improve the quality and transparency of methods and statistics reporting in our papers (please see our editorial in the May 2013 issue). Please update the Life Sciences Reporting Summary, and supplements if applicable, with any information relevant to any new experiments and upload it (as a Related Manuscript File) along with the files for your revision. If nothing in the checklist has changed, please upload the current version again.

TRANSPARENT PEER REVIEW

Nature Immunology offers a transparent peer review option for new original research manuscripts submitted from 1st December 2019. We encourage increased transparency in peer review by publishing the reviewer comments, author rebuttal letters and editorial decision letters if the authors agree. Such peer review material is made available as a supplementary peer review file. **Please state in the cover letter 'I wish to participate in transparent peer review' if you want to opt in, or 'I do not wish to participate in transparent peer review' if you don't.** Failure to state your preference will result in delays in accepting your manuscript for publication.

Please note: we allow redactions to authors' rebuttal and reviewer comments in the interest of confidentiality. If you are concerned about the release of confidential data, please let us know specifically what information you would like to have removed. Please note that we cannot incorporate redactions for any other reasons. Reviewer names will be published in the peer review files if the reviewer signed the comments to authors, or if reviewers explicitly agree to release their name. For more information, please refer to our <https://www.nature.com/documents/nr-transparent-peer-review.pdf> target="new">FAQ page.

ORCID

Nature Immunology is committed to improving transparency in authorship. As part of our efforts in this direction, we are now requesting that all authors identified as 'corresponding author' on published papers create and link their Open Researcher and Contributor Identifier (ORCID) with their account on the Manuscript Tracking System (MTS), prior to acceptance. ORCID helps the scientific community achieve unambiguous attribution of all scholarly contributions. For more information please visit <http://www.springernature.com/orcid>.

Before resubmitting the final version of the manuscript, if you are listed as a corresponding author on the manuscript, please follow the steps below to link your account on our MTS with your ORCID. If you don't have an ORCID yet, you will be able to create one in minutes. If you are not listed as a corresponding author, please ensure that the corresponding author(s) comply.

1. From the home page of the [MTS](https://mts-ni.nature.com/cgi-bin/main.plex) click on **'Modify my Springer Nature account'** under **'General tasks'**.
2. In the **'Personal profile'** tab, click on **'ORCID Create/link an Open Researcher Contributor ID(ORCID)'**. This will re-direct you to the ORCID website.
- 3a. If you already have an ORCID account, enter your ORCID email and password and click on **'Authorize'** to link your ORCID with your account on the MTS.
- 3b. If you don't yet have an ORCID, you can easily create one by providing the required information and then click on **'Authorize'**. This will link your newly created ORCID with your account on the MTS.

IMPORTANT: All authors identified as 'corresponding authors' on the manuscript must follow these instructions. Non-corresponding authors do not have to link their ORCIDs, but please note that it will not be possible to add/modify ORCIDs at proof. Thus, if they wish to have their ORCID added to the paper, they must also follow the above procedure prior to acceptance.

To support ORCID's aims, we only allow a single ORCID identifier to be attached to one account. If you have any issues attaching an ORCID identifier to your Manuscript Tracking System account, please contact the [Platform Support Helpdesk](http://platformsupport.nature.com/).

We hope that you will support this initiative and supply the required information. Should you have any query or comments, please do not hesitate to contact immunology@us.nature.com.

Nature Immunology has now transitioned to a unified Rights Collection system which will allow our Author Services team to quickly and easily collect the rights and permissions required to publish your work. Once your paper is accepted, you will receive an email in approximately 10 business days providing you with a link to complete the grant of rights. If you choose to publish Open Access, our Author Services team will also be in touch at that time regarding any additional information that may be required to arrange payment for your article.

For information regarding our different publishing models please see our [Transformative Journals](https://www.springernature.com/gp/open-research/transformational-journals) page. If you have any questions about costs, Open Access requirements, or our legal forms, please contact ASJournals@springernature.com.

In recognition of the time and expertise our reviewers provide to Nature Immunology's editorial process, we would like to formally acknowledge their contribution to the external

peer review of your manuscript entitled "IL-17D-induced keratinocyte DDX5/sIL-36R disorder drives skin inflammation". For those reviewers who give their assent, we will be publishing their names alongside the published article.

When you are ready to submit your revised manuscript, please use the URL below to submit the revised version: [REDACTED]

We hope to receive your revised manuscript in 10 days, by 9th Aug 2022. Please let us know if circumstances will delay submission beyond this time. If you have any questions please do not hesitate to contact me.

Sincerely,

Ioana Visan, Ph.D.
Senior Editor
Nature Immunology

Tel: 212-726-9207
Fax: 212-696-9752
www.nature.com/ni

Reviewer #1 (Remarks to the Author):

Extended Fig 1a-b: It is unclear why different (GO or KEGG) analysis pipelines were used to analyze the RNA-seq datasets from AD and psoriasis. Correct comparison is to use the same analysis portal when comparing across datasets. It is clear from their data that the enrichment pathways are significantly different between AD and psoriasis, this should be acknowledged.

Extended Fig 1C: Similar to the comment above, the gene expression do not correlate well between AD and psoriasis. In the introduction, the authors use these data to support the premise of this study. They should acknowledge these differences and rewrite their introduction.

Figs 1m, 2h: Numbers and percent of cells are vastly different when you compare these two figures. Gating strategies are different as well. This discrepancy needs to be explained in the text.

Extended Fig 2f: Numbers and percent of cells are vastly different when comparing psoriasis and AD like disease models. This is not discussed.

Minor:

Extended Fig 2a-d: Quantification of these data are not shown.

Fig 1I: Missing legends for this figure.

Author Rebuttal, first revision:

Point-by-point response

Reviewer #1 (Remarks to the Author):

Comment Extended Fig 1a-b: *It is unclear why different (GO or KEGG) analysis pipelines were used to analyze the RNA-seq datasets from AD and psoriasis. Correct comparison is to use the same analysis portal when comparing across datasets. It is clear from their data that the enrichment pathways are significantly different between AD and psoriasis, this should be acknowledged.*

Response: To find out the overlapping molecular characteristics between AD and psoriasis, genes that were simultaneously upregulated or downregulated in both AD and psoriasis (GSE121212) were selected out for GO or KEGG analysis. KEGG analysis was used to identify a shared pathway between AD and psoriasis, while GO analysis was used to identify molecules that exert similar functions in both AD and psoriasis (added to Extended Data Fig. 1a&b legends in page 1 of Supplementary Information). As expected, we found a similar RNA splicing pattern by the presence of shared spliceosome complexes in both AD and psoriasis, although most of enriched pathways are different. We have acknowledged this in line 64-65, page 3 in this revised version.

Comment Extended Fig 1C: *Similar to the comment above, the gene expression do not correlate well between AD and psoriasis. In the introduction, the authors use these data to support the premise of this study. They should acknowledge these differences and rewrite their introduction.*

Response: Sorry for this confusing description! Here we tried to highlight that DDX5 is the most critical factor involved in splicing changes in AD or psoriasis. We have rewritten this sentence as follow: Further analysis of the splicing signature implied that DDX5 might be potentially responsible for splicing changes in AD or psoriasis (line 65-67, page 3 in the revised version).

Comment Figs 1m, 2h: *Numbers and percent of cells are vastly different when you compare these two figures. Gating strategies are different as well. This discrepancy needs to be explained in the text.*

Response: Thanks for the reviewer's critique! We used the gating strategy for Fig. 1m to re-analyzed numbers and percent of cells in Fig. 2h (See Figure 1 attached to this point-by-point response). As shown in Fig. 2h in this revised version (line 155-156, page 7), the percent of CD45⁺ cells, eosinophils, basophils and CD4 T cells of WT mice in Fig. 2h was close to those in *Ddx5^{fl/fl}* mice in Fig. 1m. The different numbers of different cell types between *Ddx5^{fl/fl}* mice in Fig. 1m and WT mice in Fig. 2h are due to different total cell numbers used for FACS analysis (5×10^4 cells in Fig. 1m while 10^5 cells in Fig. 2h, which are shown in Y axes).

Comment Extended Fig 2f: *Numbers and percent of cells are vastly different when comparing psoriasis and AD like disease models. This is not discussed.*

Response: Accumulating evidence demonstrates that different cell types are involved in the pathogenesis of AD and psoriasis^{1,2,3}. For example, eosinophils, basophils and CD4⁺ T cells (especially Th2 cells) are major cells that play important roles in the pathogenesis of AD, while neutrophils, dendritic cells and $\gamma\delta$ T cells are main pathogenic cells for psoriasis³. DDX5 deficiency increased different levels of CCL11, CCL17 and CCL22 in keratinocytes stimulated by IL-36 (Fig. 4c), which attract eosinophils, basophils and CD4⁺ T cells in lesional skin of AD⁴. Moreover, DDX5 deficiency increased the expression of CCL20, CXCL1 and CXCL2 in keratinocytes stimulated by IL-36 (Fig. 4d), which recruit dendritic cells, $\gamma\delta$ T/Th17 cells and neutrophils into the lesional skin of psoriasis¹. Therefore, different levels of chemokines recruit different numbers and percent of different cell types in skin lesions of AD-like and psoriasis-like mice. Moreover, ears with lesions were used to collect cells for FACS analysis in Fig. 1m (line 833, page 35), Fig.2h (line 846, page 36), Fig.6f (line 919, page 39) and Extended data Fig. 8e (page 11 in Supplementary information), while dorsal skin with lesions was used to collect cells for FACS analysis in Extended Data Fig.2h (page 2 in Supplementary information). When dorsal skin was used, epidermis was peeled off to avoid noises from plaques as suggested by other groups^{5,6,7}, which leads to low percent of CD45⁺ cells as the epidermis in psoriatic lesions contained infiltrated neutrophils. Due to a word limit of Nature Immunology, we did not do further discussion in the main text, but added the necessary information in Figure legends.

Minor:

Comment Extended Fig 2a-d: *Quantification of these data are not shown.*

Response: We have added the quantification of Extended Data Fig.2b and 2d in Extended Figure 2 (page 2 in Supplementary information), but shown the quantification of Extended Data Fig.2a in Figure 2 attached to this point-by-point response due to a limit of space. For Extended Data Fig. 2c, dorsal skin cannot be quantified.

Comment Fig 1I: *Missing legends for this figure.*

Response: We have added the legend for Fig 1i (line 826, page 35).

Figures

Figure 1. Gating of eosinophils, basophils and CD4⁺ T cells in lesional skin from AD-like mice.

Figure 2. Ear thickness of *Ddx5^{fl/fl}* and *Ddx5^{Δ/kc}* mice during AD development.

References

1. Lowes, M.A., Suarez-Farinas, M. & Krueger, J.G. Immunology of psoriasis. *Annu Rev Immunol* **32**, 227-255 (2014).
2. Stander, S. Atopic Dermatitis. *N Engl J Med* **384**, 1136-1143 (2021).
3. Dainichi, T. *et al.* The epithelial immune microenvironment (EIME) in atopic dermatitis and psoriasis. *Nat Immunol* **19**, 1286-1298 (2018).

4. Gros, E., Bussmann, C., Bieber, T., Forster, I. & Novak, N. Expression of chemokines and chemokine receptors in lesional and nonlesional upper skin of patients with atopic dermatitis. *J Allergy Clin Immunol* **124**, 753-760 e751 (2009).
5. Zhang, L.J. *et al.* Age-Related Loss of Innate Immune Antimicrobial Function of Dermal Fat Is Mediated by Transforming Growth Factor Beta. *Immunity* **50**, 121-136 e125 (2019).
6. Su, Y. *et al.* Interleukin-17 receptor D constitutes an alternative receptor for interleukin-17A important in psoriasis-like skin inflammation. *Sci Immunol* **4** (2019).
7. Wang, M. *et al.* Gain-of-Function Mutation of Card14 Leads to Spontaneous Psoriasis-like Skin Inflammation through Enhanced Keratinocyte Response to IL-17A. *Immunity* **49**, 66-79 e65 (2018).

Decision Letter, second revision:

Subject: Your manuscript, NI-A32987B

Message: Our ref: NI-A32987B

2nd Sep 2022

Dear Dr. Lai,

Thank you for your patience as we've prepared the guidelines for final submission of your Nature Immunology manuscript, "IL-17D-induced keratinocyte DDX5/sIL-36R disorder drives skin inflammation" (NI-A32987B). Please carefully follow the step-by-step instructions provided in the attached file, and add a response in each row of the table to indicate the changes that you have made. Please also check and comment on any additional marked-up edits we have proposed within the text. Ensuring that each point is addressed will help to ensure that your revised manuscript can be swiftly handed over to our production team.

We would like to start working on your revised paper, with all of the requested files and forms, as soon as possible (preferably no later than September 8th). Please get in contact with us if you anticipate delays.

When you upload your final materials, please include a point-by-point response to any remaining reviewer comments and please make sure to upload your checklist.

If you have not done so already, please alert us to any related manuscripts from your group that are under consideration or in press at other journals, or are being written up for submission to other journals (see: <https://www.nature.com/nature-portfolio/editorial-policies/plagiarism#policy-on-duplicate-publication> for details).

In recognition of the time and expertise our reviewers provide to Nature Immunology's editorial process, we would like to formally acknowledge their contribution to the external peer review of your manuscript entitled "IL-17D-induced keratinocyte DDX5/sIL-36R disorder drives skin inflammation". For those reviewers who give their assent, we will be publishing their names alongside the published article.

Nature Immunology offers a Transparent Peer Review option for new original research manuscripts submitted after December 1st, 2019. As part of this initiative, we encourage our authors to support increased transparency into the peer review process by agreeing to have the reviewer comments, author rebuttal letters, and editorial decision letters

published as a Supplementary item. When you submit your final files please clearly state in your cover letter whether or not you would like to participate in this initiative. Please note that failure to state your preference will result in delays in accepting your manuscript for publication.

Cover suggestions

As you prepare your final files we encourage you to consider whether you have any images or illustrations that may be appropriate for use on the cover of Nature Immunology.

Nature Immunology has now transitioned to a unified Rights Collection system which will allow our Author Services team to quickly and easily collect the rights and permissions required to publish your work. Approximately 10 days after your paper is formally accepted, you will receive an email in providing you with a link to complete the grant of rights. If your paper is eligible for Open Access, our Author Services team will also be in touch regarding any additional information that may be required to arrange payment for your article.

Please note that *Nature Immunology* is a Transformative Journal (TJ). Authors may publish their research with us through the traditional subscription access route or make their paper immediately open access through payment of an article-processing charge (APC). Authors will not be required to make a final decision about access to their article until it has been accepted. [Find out more about Transformative Journals](https://www.springernature.com/gp/open-research/transformative-journals).

If you have any questions about costs, Open Access requirements, or our legal forms, please contact ASJournals@springernature.com.

Authors may need to take specific actions to achieve [compliance](https://www.springernature.com/gp/open-research/funding/policy-compliance-faqs) with funder and institutional open access

mandates. If your research is supported by a funder that requires immediate open access (e.g. according to [Plan S principles](https://www.springernature.com/gp/open-research/plan-s-compliance)) then you should select the gold OA route, and we will direct you to the compliant route where possible. For authors selecting the subscription publication route, the journal's standard licensing terms will need to be accepted, including [self-archiving policies](https://www.springernature.com/gp/open-research/policies/journal-policies). Those licensing terms will supersede any other terms that the author or any third party may assert apply to any version of the manuscript.

Please use the following link for uploading these materials: [REDACTED]

Best regards,

Elle Morris
Senior Editorial Assistant
Nature Immunology
Phone: 212 726 9207
Fax: 212 696 9752
E-mail: immunology@us.nature.com

On behalf of

Ioana Visan, Ph.D.
Senior Editor
Nature Immunology

Tel: 212-726-9207
Fax: 212-696-9752
www.nature.com/ni

Final Decision Letter:

Subject: Decision on Nature Immunology submission NI-A32987C

Message: In reply please quote: NI-A32987C

Dear Dr. Lai,

I am delighted to accept your manuscript entitled "IL-17D-induced inhibition of DDX5 expression in keratinocytes amplifies IL-36R-mediated skin inflammation" for publication in an upcoming issue of Nature Immunology.

Over the next few weeks, your paper will be copyedited to ensure that it conforms to Nature Immunology style. Once your paper is typeset, you will receive an email with a link to choose the appropriate publishing options for your paper and our Author Services team will be in touch regarding any additional information that may be required.

Please note that *Nature Immunology* is a Transformative Journal (TJ). Authors may publish their research with us through the traditional subscription access route or make their paper immediately open access through payment of an article-processing charge (APC). Authors will not be required to make a final decision about access to their article until it has been accepted. [Find out more about Transformative Journals](https://www.springernature.com/gp/open-research/transformative-journals).

Your paper will be published online soon after we receive your corrections and will appear in print in the next available issue. Content is published online weekly on Mondays and Thursdays, and the embargo is set at 16:00 London time (GMT)/11:00 am US Eastern time (EST) on the day of publication. Now is the time to inform your Public Relations or Press Office about your paper, as they might be interested in promoting its publication. This will allow them time to prepare an accurate and satisfactory press release. Include your manuscript tracking number (NI-A32987C) and the name of the journal, which they

will need when they contact our office.

About one week before your paper is published online, we shall be distributing a press release to news organizations worldwide, which may very well include details of your work. We are happy for your institution or funding agency to prepare its own press release, but it must mention the embargo date and Nature Immunology. Our Press Office will contact you closer to the time of publication, but if you or your Press Office have any enquiries in the meantime, please contact press@nature.com.

Also, if you have any spectacular or outstanding figures or graphics associated with your manuscript - though not necessarily included with your submission - we'd be delighted to consider them as candidates for our cover. Simply send an electronic version (accompanied by a hard copy) to us with a possible cover caption enclosed.

Please note that we encourage the authors to self-archive their manuscript (the accepted version before copy editing) in their institutional repository, and in their funders' archives, six months after publication. Nature Portfolio recognizes the efforts of funding bodies to increase access of the research they fund, and strongly encourages authors to participate in such efforts. For information about our editorial policy, including license agreement and author copyright, please visit www.nature.com/ni/about/ed_policies/index.html

Sincerely,

Ioana Visan, Ph.D.
Senior Editor
Nature Immunology

Tel: 212-726-9207
Fax: 212-696-9752
www.nature.com/ni